# Respecting Modality Gap in Post-hoc Out-of-distribution Detection with Pre-trained Vision-Language Models

Yuanwei Hu [* 1]  Bo Peng [* 1]  Yadan Luo [2]  Zhen Fang [1]  Ling Chen [1]  Jie Lu [1]

## Abstract

Out-of-distribution (OOD) detection has emerged as a popular technique to enhance the reliability of machine learning models by identifying unexpected inputs from unknown classes. Recent progress in pre-trained vision–language models (VLMs) has enabled zero-shot OOD detection without access to in-distribution (ID) training data; in this setting, existing methods commonly treat text embeddings of class names as class prototypes. In this paper, we challenge this widely adopted "text-as-prototype" paradigm by theoretically showing that off-the-shelf textual prototypes are generally misaligned with the optimal visual prototypes, yielding an intrinsic *modality gap* that cannot be eliminated by prompt engineering alone. To mitigate this gap under the post-hoc constraint, this paper presents an online pseudo-supervised framework that directly learns class prototypes in the visual feature space using unlabeled test-time data streams and soft predictions from the pre-trained VLMs. We provide theoretical guarantees for the convergence of the online optimization procedure. Extensive experiments empirically manifest that our method achieves a new state of the art across a variety of OOD detection setups. Code is available at here.

## 1. Introduction

Despite significant progress in machine learning that has enabled a broad range of classification tasks (Masana et al., 2022; Zhao et al., 2019; Caruana & Niculescu-Mizil, 2006; Peng et al., 2020; Zhu et al., 2020; 2023), most models are developed and evaluated under a *closed-world* assumption, where test samples follow the same distribution as the training data. In real-world deployments, however, models

frequently operate in *open-world* settings, where they may encounter classes unseen during training—so-called out-of-distribution (OOD) data. Such OOD inputs can undermine model reliability and, in some cases, severely degrade performance. Therefore, a dependable discriminative model should not only correctly classify in-distribution (ID) samples but also identify OOD samples as unknown. This motivates OOD detection (Lang et al., 2023; Salehi et al., 2021; Yang et al., 2021), a capability that is critical for safety in decision-sensitive applications such as autonomous driving (Huang et al., 2020), medical diagnosis (Zimmerer et al., 2022), and cybersecurity (Nguyen et al., 2022).

This paper studies *post-hoc* OOD detection, which is often more practical than learning-based approaches that require resource-intensive retraining. Earlier works (Liang et al., 2017; Liu et al., 2020; Huang et al., 2021; Sun et al., 2022; Lee et al., 2018) largely relied on single-modality information from pre-trained models. More recently, the success of contrastive language–image pre-training (CLIP) (Radford et al., 2021) has shifted attention toward extending post-hoc OOD detection from unimodal to multimodal settings.

A pioneering line of work (Esmaeilpour et al., 2022) introduces a trainable captioner to generate candidate OOD labels for matching OOD images but fails to scale to large-scale datasets, such as ImageNet-1k, that contain a large number of ID classes. On the contrary, MCM (Ming et al., 2022) treats textual features as concept prototypes for each ID class and measures OOD uncertainty via the scaled distance between an image feature and its nearest ID prototype. This approach helped establish the use of pre-trained vision–language models (VLMs) for post-hoc OOD detection. Nevertheless, MCM relies solely on textual information derived from the ID label space, leaving the text understanding capacity of VLMs under-exploited. To address this issue, NegLabel (Jiang et al., 2024) selects negative labels from in-the-wild lexical resources (e.g., WordNet (Miller, 1995)), based on similarity to the ID label space, thereby strengthening the model's ability to separate OOD from ID samples.

Despite this progress, a fundamental question remains insufficiently examined: **are text embeddings genuinely suitable surrogates for visual class prototypes in zero-shot OOD detection?** Existing CLIP-based methods implicitly

---

[1]University of Technology Sydney [2]The University of Queensland. Correspondence to: Jie Lu <Jie.Lu@uts.edu.au>.

*Proceedings of the 43rd International Conference on Machine Learning*, Seoul, South Korea. PMLR 306, 2026. Copyright 2026 by the author(s).

assume that textual representations of class names are well aligned with the optimal class prototypes in the visual feature space. While CLIP maps images and texts into a shared embedding space, recent evidence suggests that the text and vision representations learned by CLIP-based models remain distinct and separated by a clear margin (Liang et al., 2022; Shi et al.; Schrodi et al., 2024).

In this paper, we theoretically characterize an intrinsic limitation of text-as-prototype strategies in CLIP-based post-hoc OOD detection. Under an idealized supervised formulation, we derive a closed-form expression for the optimal visual class prototypes and prove that these prototypes must lie in the span of visual features. In contrast, text-derived prototypes generally include components orthogonal to this visual subspace. This mismatch yields a non-vanishing lower bound on the distance between textual prototypes and optimal visual prototypes, which we formalize as a *modality gap*. Our analysis indicates that this gap is structural and therefore cannot be removed solely through improved prompt engineering or negative-label mining.

To mitigate the modality gap, we propose a principled framework that directly learns class prototypes in the visual feature space. Specifically, we design a pseudo-supervised objective constructed from soft predictions produced by a pre-trained CLIP-based model. We further develop an online optimization procedure that incrementally updates visual prototypes using unlabeled test-time data streams, thereby respecting the post-hoc constraint. Finally, we establish theoretical guarantees on the convergence behavior of the proposed online optimization procedure. Extensive experiments show that our method achieves state-of-the-art performance. In particular, on ImageNet-1K we obtain 12.79% FPR95 and 97.75% AUROC, surpassing AdaNeg (Zhang & Zhang, 2024) by 6.13% and 1.09%, respectively.

## 2. Preliminary

**Notation.** Let $\mathcal{X}$ and $\mathcal{Y}$ be the input space and the label space, respectively. Given a random variable $Y \in \mathcal{Y}$, we write $\mathbb{P}_Y$ as the marginal distribution defined over $\mathcal{Y}$, and use $y \sim \mathbb{P}_Y$ to indicate a sample $y$ drawn from $\mathbb{P}_Y$. Considering $K$-way classification as a case study, we write $\mathcal{Y}_\mathrm{I} \triangleq \{y_1, \ldots, y_K\} \subset \mathcal{Y}$ as the *known* ID label space. The joint ID distribution $\mathbb{P}_{X_\mathrm{I} Y_\mathrm{I}}$ is a joint distribution defined over $\mathcal{X} \times \mathcal{Y}_\mathrm{I}$. During testing, there are some unknown OOD joint distributions $\mathbb{P}_{X_o Y_o}$ defined over $\mathcal{X} \times \mathcal{Y}_o$, where $\mathcal{Y}_o \subseteq \mathcal{Y} \setminus \mathcal{Y}_\mathrm{I}$ is the *unknown* OOD label space.

**Post-hoc OOD Scoring.** Existing methods (Huang et al., 2021; Sun et al., 2022; Lee et al., 2018; Wang et al., 2021; Liu et al., 2020) adopt a post-hoc strategy to detect OOD data, *i.e.,* given a pre-trained ID classification model $f$ and a scoring function $S(\cdot; f) : \mathcal{X} \to \mathbb{R}$, then $\mathbf{x}$ is detected as ID

data if and only if $S(\mathbf{x}; f) \geq \lambda$, for some given threshold $\lambda$:

$$g(\mathbf{x}) = \mathrm{ID}, \text{ if } S(\mathbf{x}; f) \geq \lambda; \text{ otherwise, } g(\mathbf{x}) = \mathrm{OOD}. \quad (1)$$

Typically, $\lambda$ is chosen to ensure a high fraction (e.g., 95%) of ID data to be correctly classified.

**CLIP-based Models** adopt a dual-stream architecture (Radford et al., 2021) with one text encoder $f_\mathcal{T}$ and one image encoder $f_\mathcal{X}$ to map inputs of two modalities into a hyperspherical space $\mathbb{S}^{d-1} \triangleq \{\mathbf{z} \in \mathbb{R}^d | \|\mathbf{z}\|_2 = 1\}$. Given any image $\mathbf{x} \in \mathcal{X}$ and any label $y \in \mathcal{Y}$, the corresponding representation can be extracted as follows:

$$\mathbf{z} = f_\mathcal{X}(\mathbf{x}) \in \mathbb{S}^{d-1}, \ \ \mathbf{r}_y = f_\mathcal{T}(\mathcal{P}(y)) \in \mathbb{S}^{d-1},$$

where $\mathcal{P}(\cdot)$ generates the text prompt for the class name associated with label $y$.

**CLIP-based OOD Detectors Studied.** CLIP-based models, which are initially proposed for zero-shot ID classification, have recently been extended to zero-shot OOD detection where there is no need to train on ID samples. The pioneering work, MCM (Ming et al., 2022), treats the prompt of ID labels as concept prototypes and measures the ID-ness of the input image by comparing the similarity between the input image and the concept prototypes in the feature space learned by CLIP-based models, i.e.,

$$\begin{aligned} S_{\mathrm{MCM}}(\mathbf{x}; f) &\triangleq \max_{y \in \mathcal{Y}_\mathrm{I}} P(y|\mathbf{x}; \mathcal{Y}_\mathrm{I}) \\ &= \frac{\max_{y \in \mathcal{Y}_\mathrm{I}} \exp(\mathbf{z} \cdot \mathbf{r}_y / \tau)}{\sum_{y \in \mathcal{Y}_\mathrm{I}} \exp(\mathbf{z} \cdot \mathbf{r}_y / \tau)}, \end{aligned} \quad (2)$$

where $\tau > 0$ is a temperature hyper-parameter. Built upon this, NegLabel (Jiang et al., 2024) introduces a $L$-sized set of negative labels[1] $\mathcal{Y}_{\mathrm{neg}} \triangleq \{y_{K+1}, \ldots, y_{K+L}\}$ to formulate the OOD scoring function of $\mathbf{x}$ as the model's prediction confidence that $\mathbf{x}$ belongs to $\mathcal{Y}_\mathrm{I}$, i.e.,

$$\begin{aligned} S_{\mathrm{NegLabel}}(\mathbf{x}; f) &\triangleq \sum_{y \in \mathcal{Y}_\mathrm{I}} P(y|\mathbf{x}; \mathcal{Y}_\mathrm{I} \cup \mathcal{Y}_{\mathrm{neg}}) \\ &= \frac{\sum_{y \in \mathcal{Y}_\mathrm{I}} \exp(\mathbf{z} \cdot \mathbf{r}_y / \tau)}{\sum_{y \in \mathcal{Y}_\mathrm{I} \cup \mathcal{Y}_{\mathrm{neg}}} \exp(\mathbf{z} \cdot \mathbf{r}_y / \tau)}. \end{aligned} \quad (3)$$

## 3. Motivation

Despite the empirical success of CLIP-based methods for post-hoc OOD detection, we argue that their potential remains fundamentally under-exploited. The key limitation stems from the absence of training data in the zero-shot setting. As a result, existing approaches typically construct class prototypes directly from the text embeddings of class names. While computationally convenient, this practice

---

[1] By definition, *negative* labels are those semantically *irrelevant/dissimilar* to *all* ID labels.

implicitly assumes that textual prototypes are well-aligned with their optimal counterparts in the visual feature space.

In this work, we argue that this assumption is generally invalid. Even though CLIP embeds images and texts into a shared hyperspherical space, text-derived prototypes often exhibit a systematic mismatch with optimal visual prototypes. We refer to this discrepancy as the *modality gap*. To make this precise, we first analyze an idealized supervised setting where optimal visual prototypes can be characterized explicitly. This analysis provides a theoretical reference point for understanding the intrinsic limitations of text-only prototypes in zero-shot OOD detection.

Without loss of generality, let $\mathcal{D} = \{(\mathbf{x}_i, \hat{y}_i)\}_{i=1}^N \sim \mathbb{P}_{X\hat{Y}}^N$ where $\hat{\mathcal{Y}} \subseteq \mathcal{Y}$ denotes a label space of interest with cardinality $|\hat{\mathcal{Y}}| = M$. The optimal class prototypes w.r.t. $\hat{\mathcal{Y}}$ can be learned by minimizing the following supervised objective:

$$\mathcal{L}(\mathbf{W}; \hat{\mathcal{Y}}) = -\sum_{i=1}^N \log \frac{\exp(\mathbf{z}_i \cdot \mathbf{w}_{\hat{y}_i}/\kappa)}{\sum_{\hat{y}\in\hat{y}} \exp(\mathbf{z}_i \cdot \mathbf{w}_{\hat{y}}/\kappa)}, \quad (4)$$

where $\kappa > 0$ is another temperature hyper-parameter and $\mathbf{W} = [\mathbf{w}_{\hat{y}}]_{\hat{y}\in\hat{y}}$ with $\mathbf{w}_{\hat{y}} \in \mathbb{S}^{d-1}$ is the class prototype associated with label $\hat{y}$ in the visual feature space.

**Theorem 1.** *Let $\mathbf{W}^* = \arg\min_{\mathbf{W}} \mathcal{L}(\mathbf{W}; \hat{\mathcal{Y}})$, then we have*

$$\mathbf{w}_{\hat{y}}^* = \ell_2 \left( \sum_{(\mathbf{x}_i, \hat{y}_i)\in\mathcal{D}_{\hat{y}}} (1 - \pi_{i\hat{y}})\mathbf{z}_i - \sum_{(\mathbf{x}_i, \hat{y}_i)\in\mathcal{D}\setminus\mathcal{D}_{\hat{y}}} \pi_{i\hat{y}}\mathbf{z}_i \right),$$

*where $\mathcal{D}_{\hat{y}} = \{(\mathbf{x}_i, \hat{y}_i) \in \mathcal{D} : \hat{y}_i = \hat{y}\}$ and*

$$\pi_{i\hat{y}} = \frac{\exp(\mathbf{z}_i \cdot \mathbf{w}_{\hat{y}}^*/\kappa)}{\sum_{y\in\hat{y}} \exp(\mathbf{z}_i \cdot \mathbf{w}_y^*/\kappa)}.$$

*Proof.* See Appendix B for the proof. □

According to Theorem 1, one may conclude that each optimal class prototype $\mathbf{w}_{\hat{y}}^*$ should entirely lie in the visual feature space since it can be expressed as a linear combination of samples from that space. This conclusion will be central to our following analysis of the modality gap.

**Theorem 2.** *Let $\mathbf{R} = [\mathbf{r}_{\hat{y}}]_{\hat{y}\in\hat{y}}$ where $\mathbf{r}_{\hat{y}} \in \mathbb{S}^{d-1}$ denotes the class prototype associated with label $\hat{y}$ in the textual feature space. Let $Proj_{\mathcal{S}}(\cdot)$ and $Proj_{\mathcal{S}^\perp}(\cdot)$ denote the orthogonal projectors onto the visual feature subspace $\mathcal{S} = span\{\mathbf{z}_1, \ldots, \mathbf{z}_N\}$ and its orthogonal complement $\mathcal{S}^\perp$, respectively, then we have*

$$\|\mathbf{R} - \mathbf{W}^*\|_F^2 \geq \sum_{\hat{y}\in\hat{\mathcal{Y}}} 2\left(1 - \|Proj_{\mathcal{S}}(\mathbf{r}_{\hat{y}})\|_2\right). \quad (5)$$

*Equivalently,*

$$\|\mathbf{R} - \mathbf{W}^*\|_F^2 \geq \sum_{\hat{y}\in\hat{\mathcal{Y}}} 2\left(1 - \sqrt{1 - \|Proj_{\mathcal{S}^\perp}(\mathbf{r}_{\hat{y}})\|_2^2}\right). \quad (6)$$

*Proof.* See Appendix C for the proof □

**Remark.** Theorem 2 establishes that the gap between textual prototypes and optimal visual prototypes is unavoidable whenever textual prototypes contain components outside the visual feature subspace $\mathcal{S}$. Since $\mathbf{w}_{\hat{y}}^* \in \mathcal{S}$ by Theorem 1, only $Proj_{\mathcal{S}}(\mathbf{r}_{\hat{y}})$ can align with $\mathbf{w}_{\hat{y}}^*$, while $Proj_{\mathcal{S}^\perp}(\mathbf{r}_{\hat{y}})$ necessarily contributes to the mismatch. Hence, the lower bound in Theorem 2 vanishes only when each textual prototype lies entirely in $\mathcal{S}$. As a result, using text embeddings as class prototypes *without visual calibration* is generally sub-optimal for recovering the optimal visual prototypes.

## 4. Methodology

### 4.1. An online pseudo-supervised learning framework

Albeit theoretically appealing, it is worth noting that recovering $\mathbf{W}^*$ by minimizing Eq. (4) is technically infeasible in the zero-shot setting where the labeled training dataset $\mathcal{D}$ is unavailable. While one can have access to unlabeled samples during testing, the absence of ground-truth labels precludes direct optimization of the supervised objective in Eq. (4). Fortunately, CLIP-based models exhibit strong zero-shot transfer performance, suggesting that their predictions can serve as a surrogate supervision signal. We thus replace hard ground-truth labels with soft pseudo-labels predicted by CLIP-based models, directly yielding a pseudo-supervised objective $\hat{\mathcal{L}}(\mathbf{W}) = \sum_{i=1}^N \hat{\ell}(\mathbf{W}; \mathbf{x}_i, \hat{\mathcal{Y}})$, where

$$\hat{\ell}(\mathbf{W}; \mathbf{x}_i, \hat{\mathcal{Y}}) \triangleq$$
$$-\sum_{\hat{y}\in\hat{y}} P(\hat{y}|\mathbf{x}_i; \hat{\mathcal{Y}}) \log \frac{\exp(\mathbf{z}_i \cdot \mathbf{w}_{\hat{y}}/\kappa)}{\sum_{y\in\hat{y}} \exp(\mathbf{z}_i \cdot \mathbf{w}_y/\kappa)}. \quad (7)$$

Given that only a single test sample will be received at each iteration, we propose to update $\mathbf{W}$ in an online manner. In particular, when the $i$-th test sample $\mathbf{x}_i$ arrives, we have

$$\mathbf{w}_{\hat{y}}^{(i)} = \ell_2 \left( \mathbf{w}_{\hat{y}}^{(i-1)} - \eta_i \frac{\partial \hat{\ell}(\mathbf{W}^{(i-1)}; \mathbf{x}_i, \hat{\mathcal{Y}})}{\partial \mathbf{w}_{\hat{y}}^{(i-1)}} \right), \forall \hat{y} \in \hat{\mathcal{Y}} \quad (8)$$

where $\eta_i = \rho/\sqrt{i}$ is a step size with $\rho > 0$ as a hyperparameter. For completeness, Theorem 3 provides a theoretical guarantee for the convergence of our proposed online optimization in Eq. (8).

**Theorem 3.** *Assume there exists $\epsilon > 0$ such that, for all possible $\mathbf{W}$ and $\mathbf{x}$, $\left\|\nabla_{\mathbf{W}} \hat{\ell}(\mathbf{W}; \mathbf{x}, \hat{\mathcal{Y}})\right\|_F \leq \epsilon$. Let $\{\mathbf{W}^{(i)}\}_{i=0}^{N-1}$ be produced by Eq. (8) with $\eta_i = \rho/\sqrt{i}$ and $\rho > 0$. Then the cumulative regret is upper-bounded by*

$$\sum_{i=1}^N \hat{\ell}(\mathbf{W}^{(i-1)}; \mathbf{x}_i, \hat{\mathcal{Y}}) - \min_{\mathbf{W}} \hat{\mathcal{L}}(\mathbf{W}) \leq \mathcal{O}(\sqrt{N}), \quad (9)$$

*where we use $\mathcal{O}(\cdot)$ to hide universal constants.*

*Proof.* See Appendix D for the proof ☐

### 4.2. Online Prototype Learning for OOD Detection

To realize the idea, we, motivated by NegLabel (Jiang et al., 2024), begin by extending the ID label space to include $L$ additional placeholders. This is essentially equivalent to grouping potential target OOD data into $L$ OOD pseudo-classes (clusters). For clarity, we denote the corresponding set of OOD pseudo-labels as $\hat{\mathcal{Y}}_o = \{\hat{y}_1, \ldots, \hat{y}_L\}$.

Due to the lack of explicit OOD information, NegLabel uses text features of negative labels as OOD pseudo-class prototypes such that $\hat{\mathcal{Y}}_o = \mathcal{Y}_{neg}$. However, this strategy suffers not only from the intra-modality discrepancy between negative labels and the true underlying OOD categories, but also from the inter-modality gap between text-derived prototypes and optimal visual prototypes, as discussed in Section 3.

In contrast, as we show later, we directly learn the prototypes of both ID classes and OOD pseudo-classes in the visual feature space via the proposed online optimization framework introduced in Section 4.1.

At high level, our key idea is to update ID class prototypes if an incoming test sample is classified as ID, and to update OOD pseudo-class prototypes otherwise. Since no human annotations indicating ID or OOD are available at test time, we introduce a hard thresholding rule based on the off-the-shelf score $S_{NegLabel}(\mathbf{x}_i; f)$ in Eq. (3) to identify positive and negative samples from test data streams:

$$\begin{aligned} \text{Positive} &: S_{NegLabel}(\mathbf{x}_i; f) \geq \beta, \\ \text{Negative} &: S_{NegLabel}(\mathbf{x}_i; f) \leq 1 - \beta, \end{aligned} \quad (10)$$

where $\beta \in (0.5, 1]$. We note that, with Eq. (10), test samples whose scores fall within $(1 - \beta, \beta)$ are treated as uncertain and are not used to update any prototypes.

Given the $i$-th test sample $\mathbf{x}_i$ ($i = 1, 2, \ldots$) to arrive, we denote by $c_i^+$ and $c_i^-$ to represent the cumulative number of positive and negative samples we have ever seen so far respectively such that

$$\begin{aligned} c_i^+ &= c_{i-1}^+ + \mathbb{1}\left(S_{NegLabel}(\mathbf{x}_i; f) \geq \beta\right), \\ c_i^- &= c_{i-1}^- + \mathbb{1}\left(S_{NegLabel}(\mathbf{x}_i; f) \leq 1 - \beta\right), \end{aligned} \quad (11)$$

with $c_0^+ = c_0^- = 0$ and $\mathbb{1}(\cdot)$ as an indicator function.

For each $y \in \mathcal{Y}_I$, the ID class prototypes $\mathbf{w}_y$ is updated as

$$\mathbf{w}_y^{(i)} = \begin{cases} \mathbf{w}_y^{(i-1)} & \text{if } S_{NegLabel}(\mathbf{x}_i; f) < \beta, \\ \ell_2\left(\mathbf{w}_y^{(i-1)} - \eta_i^+ \frac{\partial \hat{\ell}(\mathbf{W}^{(i-1)}; \mathbf{x}_i, \mathcal{Y}_I)}{\partial \mathbf{w}_y^{(i-1)}}\right) & \text{otherwise,} \end{cases} \quad (12)$$

where the step size is $\eta_i^+ = \rho/\sqrt{c_i^+}$.

---

**Algorithm 1** Online Prototype Learning for OOD Detection

1: **Input:** Pre-trained CLIP-based model $f$, Prompt template $\mathcal{P}(\cdot)$, ID labels $\mathcal{Y}_I = \{y_1, \ldots, y_K\}$, OOD pseudo-labels $\hat{\mathcal{Y}}_o = \{\hat{y}_1, \ldots, \hat{y}_L\}$.
2: Initialize $c_0^+ = c_0^- = 0$
3: Initialize ID and OOD class prototypes via Eq. (15)
4: **while** the $i$-th test sample $\mathbf{x}_i$ arrives **do**
5:     Update $c_i^+$ and $c_i^-$ via Eq. (11)
6:     Update $\mathbf{w}_y^{(i)}$ via Eq. (12), $\forall y \in \mathcal{Y}_I$
7:     Update $\mathbf{w}_y^{(i)}$ via Eq. (13), $\forall y \in \hat{\mathcal{Y}}_o$
8:     Compute $S_{ours}(\mathbf{x}_i; f)$ via Eq. (14)
9: **end while**

---

Similarly, for each $y \in \hat{\mathcal{Y}}_o$, we can update the OOD pseudo-class prototype $\mathbf{w}_y$ as

$$\mathbf{w}_y^{(i)} = \begin{cases} \mathbf{w}_y^{(i-1)} & \text{if } S_{NegLabel}(\mathbf{x}_i; f) > 1 - \beta, \\ \ell_2\left(\mathbf{w}_y^{(i-1)} - \eta_i^- \frac{\partial \hat{\ell}(\mathbf{W}^{(i-1)}; \mathbf{x}_i, \hat{\mathcal{Y}}_o)}{\partial \mathbf{w}_y^{(i-1)}}\right) & \text{otherwise.} \end{cases} \quad (13)$$

where the step size is $\eta_i^- = \rho/\sqrt{c_i^-}$.

Based on the updated prototypes, we define the online OOD scoring function for $\mathbf{x}_i$ as

$$S_{ours}(\mathbf{x}_i; f) = \frac{\sum_{y \in \mathcal{Y}_I} \exp(\mathbf{z}_i \cdot \mathbf{w}_y^{(i)}/\tau)}{\sum_{y \in \mathcal{Y}_I \cup \hat{\mathcal{Y}}_o} \exp(\mathbf{z}_i \cdot \mathbf{w}_y^{(i)}/\tau)}. \quad (14)$$

where the prototypes are initialized as

$$\begin{aligned} \mathbf{w}_{y_i}^{(0)} &= f_\mathcal{T}\left(\mathcal{P}(y_i)\right), \quad \forall i = 1, \ldots, K, \\ \mathbf{w}_{\hat{y}_i}^{(0)} &= f_\mathcal{T}\left(\mathcal{P}(y_{k+i})\right), \quad \forall i = 1, \ldots, L. \end{aligned} \quad (15)$$

For clarity, we summarize the complete algorithm in Algorithm 1. Finally, we note that the proposed framework can also be applied to the ID-label-only case like MCM; detailed discussions are deferred to Appendix E.

### 4.3. Discussion

We recently found that AdaNeg (Zhang & Zhang, 2024) also performs online updates of class prototypes using unlabeled test data streams. However, our method fundamentally differs from AdaNeg in problem formulation. AdaNeg is motivated by an empirical issue, i.e., negative text labels can be semantically misaligned with the target OOD distribution, and addresses it by constructing adaptive negative prototypes from cached test-image features. While effective, this strategy remains a form of prototype engineering within the standard CLIP paradigm: it insists on treating text embeddings as valid anchors and focuses on refining negative prototypes, rather than questioning the validity of text-derived prototypes more generally.

*Table 1.* OOD detection results on ImageNet-1K, where a VIT B/16 CLIP encoder is adopted. ↑ indicates larger values are better and vice versa. The best results in the last two columns are shown in bold. Full results of our method are provided in Table 13. †: **It can be checked that the final score for OOD detection linearly combines the AdaNeg score and the NegLabel score.**

| Dataset | iNaturalist | | SUN | | Places | | Textures | | Average | |
|---|---|---|---|---|---|---|---|---|---|---|
| Metric | AUROC↑ | FPR95↓ | AUROC↑ | FPR95↓ | AUROC↑ | FPR95↓ | AUROC↑ | FPR95↓ | AUROC↑ | FPR95↓ |
| **Methods requiring training (or fine-tuning)** | | | | | | | | | | |
| MSP | 87.44 | 58.36 | 79.73 | 73.72 | 79.67 | 74.41 | 79.69 | 71.93 | 81.63 | 69.61 |
| ODIN | 94.65 | 30.22 | 87.17 | 54.04 | 85.54 | 55.06 | 87.85 | 51.67 | 88.80 | 47.75 |
| Energy | 95.33 | 26.12 | 92.66 | 35.97 | 91.41 | 39.87 | 86.76 | 57.61 | 91.54 | 39.89 |
| GradNorm | 72.56 | 81.50 | 72.86 | 82.00 | 73.70 | 80.41 | 70.26 | 79.36 | 72.35 | 80.82 |
| ViM | 93.16 | 32.19 | 87.19 | 54.01 | 83.75 | 60.67 | 87.18 | 53.94 | 87.82 | 50.20 |
| KNN | 94.52 | 29.17 | 92.67 | 35.62 | 91.02 | 39.61 | 85.67 | 64.35 | 90.97 | 42.19 |
| VOS | 94.62 | 28.99 | 92.57 | 36.88 | 91.23 | 38.39 | 86.33 | 61.02 | 91.19 | 41.32 |
| NPOS | 96.19 | 16.58 | 90.44 | 43.77 | 89.44 | 45.27 | 88.80 | 46.12 | 91.22 | 37.93 |
| LSN | 95.83 | 21.56 | 94.35 | 26.32 | 91.25 | 34.48 | 90.42 | 38.54 | 92.96 | 30.22 |
| CLIPN | 95.27 | 23.94 | 93.93 | 26.17 | 92.28 | 33.45 | 90.93 | 40.83 | 93.10 | 31.10 |
| LoCoOp | 96.86 | 16.05 | 95.07 | 23.44 | 91.98 | 32.87 | 90.19 | 42.28 | 93.52 | 28.66 |
| LAPT | 99.63 | 1.16 | 96.01 | 19.12 | 92.01 | 33.01 | 91.06 | 40.32 | 94.68 | 23.40 |
| NegPrompt | 98.73 | 6.32 | 95.55 | 22.89 | 93.34 | 27.60 | 91.60 | 35.21 | 94.81 | 23.01 |
| **Zero-Shot Training-free Methods** | | | | | | | | | | |
| Mahalanobis | 55.89 | 99.33 | 59.94 | 99.41 | 65.96 | 98.54 | 64.23 | 98.46 | 61.50 | 98.94 |
| Energy | 85.09 | 81.08 | 84.24 | 79.02 | 83.38 | 75.08 | 65.56 | 93.65 | 79.57 | 82.21 |
| ZOC | 86.09 | 87.30 | 81.20 | 81.51 | 83.39 | 73.06 | 76.46 | 98.90 | 81.79 | 85.19 |
| MCM | 94.59 | 32.20 | 92.25 | 38.80 | 90.31 | 46.20 | 86.12 | 58.50 | 90.82 | 43.93 |
| GL-MCM | 96.71 | 15.16 | 93.41 | 29.16 | 90.37 | 37.07 | 83.11 | 55.85 | 90.90 | 35.06 |
| NegLabel | 99.49 | 1.91 | 95.49 | 20.53 | 91.64 | 35.59 | 90.22 | 43.56 | 94.21 | 25.40 |
| DNM | 99.51 | 1.78 | 95.88 | 16.90 | 91.95 | 32.13 | 90.72 | 38.67 | 94.52 | 22.33 |
| InfoOOD | 99.70 | 1.04 | 96.16 | 16.06 | 93.37 | 26.92 | 91.01 | 40.78 | 95.07 | 21.20 |
| CSP | 99.60 | 1.54 | 96.66 | 13.66 | 92.90 | 29.32 | 93.86 | 25.52 | 95.76 | 17.51 |
| AdaNeg+NegLabel† | 99.71 | 0.59 | 97.44 | 9.50 | 94.55 | 34.34 | 94.93 | 31.27 | 96.66 | 18.92 |
| Ours (Mean) | 99.88 | 0.50 | 98.94 | 5.00 | 95.17 | 28.55 | 97.01 | 17.12 | **97.75** | **12.79** |
| Ours (Std) | 0.04 | 0.03 | 0.04 | 0.27 | 0.16 | 0.59 | 0.27 | 0.77 | 0.07 | 0.31 |

In contrast, our work targets a structural limitation in CLIP-based post-hoc OOD detection: directly using text embeddings as class prototypes is theoretically unjustified due to an intrinsic modality gap between text-derived prototypes and the optimal visual prototypes. This failure mode is independent of negative-label mining or memory heuristics and therefore cannot be addressed by negative-prototype adaptation alone. We instead replace text-as-prototype with a principled online pseudo-supervised optimization that calibrates prototypes in the visual feature space from unlabeled test streams and comes with convergence guarantees, yielding a more general and theoretically grounded test-time prototype calibration framework. Empirically, we show in Section 5 that these theoretical advantages translate into consistent performance gains across standard benchmarks.

## 5. Experiments

**Baselines.** We compare our method with MSP (Hendrycks & Gimpel, 2016), ODIN (Liang et al., 2017), Energy (Liu et al., 2020), Gradnorm (Huang et al., 2021), Vim (Wang et al., 2022a), KNN (Sun et al., 2022), VOS (Du et al., 2022), NPOS (Tao et al., 2023), ZOC (Esmaeilpour et al.,

2022), CLIPN (Wang et al., 2023a), LoCoOp (Miyai et al., 2024), LSN (Nie et al., 2024), LAPT (Zhang et al., 2025), NegPrompt (Li et al., 2024), Mahalanobis (Lee et al., 2018), MCM (Ming et al., 2022), NegLabel (Jiang et al., 2024), GL-MCM (Miyai et al., 2023), DNM (Peng et al., 2026c), InfoOOD (Peng et al., 2026b), AdaNeg (Zhang & Zhang, 2024) and CSP (Chen et al., 2024).

**Implementation Details.** Unless otherwise specified, we employ CLIP-B/16 for zero-shot OOD detection. Following prior works (Jiang et al., 2024; Zhang & Zhang, 2024), we adopt the text prompt of 'The nice <label>.' and use the same $L = 10000$ negative labels selected by Jiang et al. (2024). Regarding hyper-parameters, we fix $\tau = 0.01$, $\kappa = 0.05$, $\rho = 0.1$ and $\beta = 0.95$ for all experiments. *The reported results of our method are averaged over 10 independent runs.*

### 5.1. Main Results

**Evaluation on ImageNet Benchmark.** Following prior work (Ming et al., 2022; Jiang et al., 2024; Zhang & Zhang, 2024), we evaluate our method on the popular ImageNet-1K benchmark (Deng et al., 2009) in Table 1, where the vali-

*Table 2.* OOD detection results on the OpenOOD benchmark, where ImageNet-1K is adopted as ID. ↑ indicates larger values are better and vice versa. The best results are shown in bold. Full results are provided in Table 14.

| Methods | FPR95↓ | | AUROC↑ | |
|---|---|---|---|---|
| | Near-OOD | Far-OOD | Near-OOD | Far-OOD |
| *Methods requiring training (or fine-tuning)* | | | | |
| GEN | – | – | 78.97 | 90.98 |
| AugMix + ReAct | – | – | 79.94 | 93.70 |
| RMDS | – | – | 80.09 | 92.60 |
| SCALE | – | – | 81.36 | 96.53 |
| AugMix + ASH | – | – | 82.16 | 96.05 |
| LAPT | 58.94 | 24.86 | 82.63 | 94.26 |
| *Zero-Shot Training-free Methods* | | | | |
| MCM | 79.02 | 68.54 | 60.11 | 84.77 |
| NegLabel | 69.45 | 23.73 | 75.18 | 94.85 |
| AdaNeg+Neglabel | 67.51 | 17.31 | 76.70 | 96.43 |
| **Ours** | **52.86** | **13.22** | **85.80** | **97.67** |

*Table 3.* OOD detection results on the OpenOOD benchmark, where CIFAR-10 is adopted as ID. ↑ indicates larger values are better and vice versa. The best results are shown in bold. Full results are provided in Table 15.

| Methods | FPR95↓ | | AUROC↑ | |
|---|---|---|---|---|
| | Near-OOD | Far-OOD | Near-OOD | Far-OOD |
| *Methods requiring training (or fine-tuning)* | | | | |
| PixMix + KNN | – | – | 93.10 | 95.94 |
| OE + MSP | – | – | 94.82 | 96.00 |
| PixMix + RotPred | – | – | 94.86 | 98.18 |
| *Zero-Shot Training-free Methods* | | | | |
| MCM | 30.86 | 17.99 | 91.92 | 95.54 |
| NegLabel | 28.75 | 6.60 | 94.58 | 98.39 |
| AdaNeg+Neglabel | 20.40 | 2.79 | 94.78 | 99.26 |
| **Ours** | **16.38** | **2.25** | **95.72** | **99.83** |

dation set of ImageNet-1K is designated as the ID dataset while iNaturalist (Van Horn et al., 2018), SUN (Xiao et al., 2010), Places365 (Zhou et al., 2017), and Textures (Cimpoi et al., 2014) are considered as OOD datasets. At test time, all images are resized to $224 \times 224$. The baselines range from traditional fine-tuning methods and advanced post-hoc methods from pre-trained VLMs like CLIP. Our method consistently outperforms existing baselines, which highlights its superiority in the zero-shot setting. The significant improvements demonstrate the effectiveness of our method in zero-shot scenarios. Table 2 shows that the empirical advantages of our method over the state-of-the-art still hold under the OpenOOD setup (Yang et al., 2022), where Near-OOD datasets are SSB-hard (Vaze et al., 2021) and NINCO (Bitterwolf et al., 2023) while Far-OOD datasets are iNaturalist (Van Horn et al., 2018), Textures (Cimpoi et al., 2014), and OpenImage-O (Wang et al., 2022b).

**Evaluation on CIFAR Benchmarks** We further conduct experiments on CIFAR benchmarks (Krizhevsky et al., 2009) under the OpenOOD setup. To be specific, with CIFAR-10/CIFAR-100 as ID dataset, Near-OOD datasets

*Table 4.* OOD detection results on the OpenOOD benchmark, where CIFAR-100 is adopted as ID. ↑ indicates larger values are better and vice versa. The best results are shown in bold. Full results are provided in Table 16.

| Metric | FPR95↓ | | AUROC↑ | |
|---|---|---|---|---|
| | Near-OOD | Far-OOD | Near-OOD | Far-OOD |
| *Methods requiring training (or fine-tuning)* | | | | |
| GEN | – | – | 81.31 | 79.68 |
| VOS + EBO | – | – | 80.93 | 81.32 |
| SCALE | – | – | 80.99 | 81.42 |
| OE + MSP | – | – | 88.30 | 81.41 |
| *Zero-Shot Training-free Methods* | | | | |
| MCM | 75.20 | 59.32 | 71.00 | 76.00 |
| NegLabel | 71.44 | 40.92 | 70.58 | 89.68 |
| AdaNeg+Neglabel | 59.07 | 29.35 | 84.62 | 95.25 |
| **Ours** | **45.70** | **21.97** | **89.36** | **96.90** |

are CIFAR-100/CIFAR-10 and Tiny-ImageNet (Le & Yang, 2015) while Far-OOD datasets are MNIST (LeCun et al., 2010), SVHN (Netzer et al., 2011), Texture (Cimpoi et al., 2014) and Places (Zhou et al., 2017). Experiment results in Table 3 and Table 4 show that our method achieves consistent and notable improvements on both CIFAR-10 and CIFAR-20, demonstrating the generalization of our method.

### 5.2. Ablation Study

**Hyper-parameters.** We evaluate the hyper-parameters most essential to our algorithm design, including $\kappa$ in Eq. (7), $\beta$ in Eq. (10), and $\rho$ in Eq. (12) and Eq. (13). The corresponding results are displayed in Figure 1, where one can find that our method is not sensitive to hyper-parameters.

**Visual Encoders.** In principle, our method is generic to the choice of visual encoder. We evaluate our method with different visual encoder architectures, including ResNet-50, ViT-B/32 and ViT-L/14, and report the corresponding OOD detection results in Table 5. It can be seen that the performance can be improved by more powerful visual encoders. While our method consistently outperforms the most recent AdaNeg regardless of the backbone architecture used, indicating better generalization.

**Input Resolution.** In principle, our method is generic to the input resolution. We evaluate our method with a larger input size, i.e., 336×336, and report the corresponding OOD detection results in Table 6. On the one hand, the performance of OOD detection can be enhanced by a larger input size. On the other hand, our method consistently outperforms the most recent AdaNeg+NegLabel regardless of input resolution, which implies the better generalization of our method.

*Due to space limitation, more ablation studies can be found in Appendix G while analysis on time and memory complexity can be found in Appendix H*

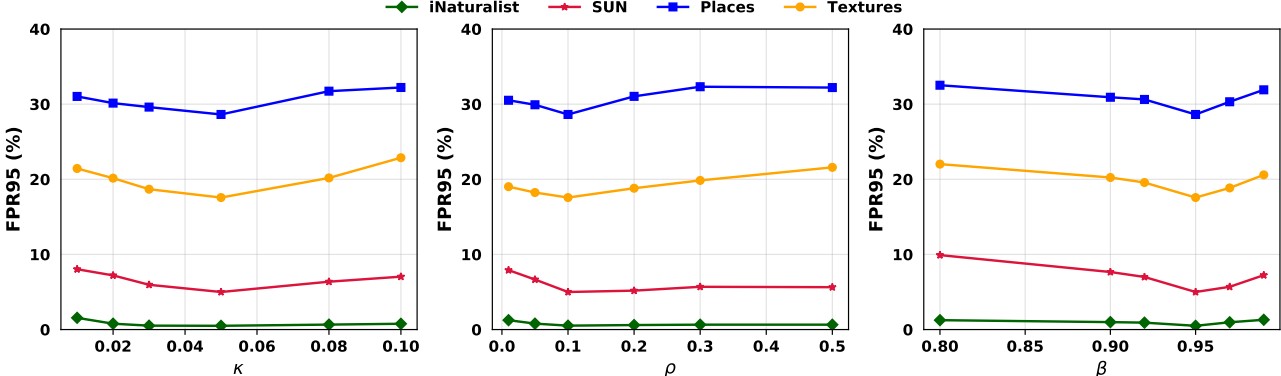

*Figure 1.* Ablation study on ImageNet-1k w.r.t. hyperparameter $\kappa$ (left), $\rho$ (middle) and $\beta$ (right).

*Table 5.* OOD detection results with different CLIP architectures on ImageNet-1K as ID. ↑ indicates larger values are better and vice versa. The best results in the last two columns are shown in bold.

| Backbone | Method | iNaturalist | | SUN | | Places | | Textures | | Average | |
|---|---|---|---|---|---|---|---|---|---|---|---|
| | | AUROC↑ | FPR95↓ | AUROC↑ | FPR95↓ | AUROC↑ | FPR95↓ | AUROC↑ | FPR95↓ | AUROC↑ | FPR95↓ |
| ResNet50 | Neglabel | 99.24 | 2.88 | 94.54 | 26.51 | 89.72 | 42.60 | 88.40 | 50.80 | 92.97 | 30.70 |
| | AdaNeg+Neglabel | 99.58 | 1.18 | 97.37 | 10.56 | 93.84 | 43.19 | 94.18 | 35.00 | 96.24 | 22.48 |
| | Ours | 99.48 | 1.11 | 98.04 | 6.05 | 94.68 | 32.61 | 96.15 | 23.76 | **97.09** | **15.88** |
| ViT-B/32 | Neglabel | 99.11 | 3.72 | 95.27 | 22.48 | 91.72 | 34.94 | 88.57 | 50.51 | 93.67 | 27.92 |
| | AdaNeg+Neglabel | 99.59 | 1.02 | 97.53 | 9.63 | 93.99 | 38.45 | 94.21 | 37.92 | 96.33 | 21.76 |
| | Ours | 99.51 | 0.84 | 98.83 | 5.33 | 94.68 | 31.73 | 96.08 | 22.54 | **97.28** | **15.11** |
| ViT-L/14 | Neglabel | 99.53 | 1.77 | 95.63 | 22.33 | 93.01 | 32.22 | 89.71 | 42.92 | 94.47 | 24.81 |
| | AdaNeg+Neglabel | 99.74 | 0.56 | 97.79 | 9.34 | 94.96 | 34.07 | 95.01 | 30.06 | 96.87 | 18.50 |
| | Ours | 99.92 | 0.35 | 99.06 | 4.17 | 96.49 | 27.49 | 97.90 | 16.45 | **98.34** | **12.12** |

## 5.3. Extensions

**Domain-generalizable OOD Detection.** We investigate domain generalizable OOD detection scenarios, where there exist domain shifts in ID data. With ImageNet-1K as a case study, we, following Jiang et al. (2024), consider ImageNet-A (Hendrycks et al., 2021b), ImageNet-R (Hendrycks et al., 2021a) and ImageNet-S (Wang et al., 2019) as ID data respectively. The experiment results on four OOD datasets are shown in Table 7. Our method consistently achieves the state-of-the-art performance, which implies the robustness of our method to domain shift.

**OOD Detection with Learned Prompt.** While this paper, following (Jiang et al., 2024), to use a pre-defined prompts for both negative and ID labels, we show in Table 8 that our method can be made stronger with learned prompts by LAPT (Zhang et al., 2025).

## 6. Related Work

### 6.1. Traditional Out-of-distribution Detection

The popularity of OOD detection is motivated by the empirical observation (Nguyen et al., 2015) that neural networks tend to be over-confident in OOD data. One line of

work performs OOD detection by devising post-hoc scoring functions, including confidence-based methods (Hendrycks et al., 2019; Ming et al., 2022; Zhang & Xiang, 2023), energy-based methods (Liu et al., 2020; Wang et al., 2021), distance-based approaches (Peng et al., 2025b; Lee et al., 2018; Sun et al., 2022; Zhang et al., 2024; Sastry & Oore, 2020; Morteza & Li, 2022; Peng et al., 2024), gradient-based approaches (Huang et al., 2021; Liao et al., 2026b), and Bayesian approaches (Kristiadi et al., 2020; Malinin & Gales, 2019). Another line of work addresses OOD detection by fine-tuning a pre-trained discrimination model with training-time regularizations that help the model learn ID/OOD discrepancy following the guideline of outlier exposure (Hendrycks et al., 2018). For instance, the discriminative model is regularized to produce lower confidence (Lee et al., 2017; Malinin & Gales, 2018; Wang et al., 2023b), smaller feature magnitudes (Liu et al., 2020) or higher energy (Dhamija et al., 2018) for outlier points. More recently, some works have considered a practical scenario where the auxiliary outliers can be arbitrarily different from the real OOD data, therefore distributionally augmenting the observed OOD data. Besides, the given OOD samples tend to include unlabelled ID counterparts (Katz-Samuels et al., 2022). Due to this, WOOD (Katz-Samuels et al., 2022) formulates learning with noisy OOD samples as a constrained

*Table 6.* OOD detection results with a larger input resolution on ImageNet-1k as ID, where a VIT L/14 CLIP encoder is adopted. ↑ indicates larger values are better and vice versa. The best results in the last two columns are shown in bold.

| Resolution | Method | iNaturalist | | SUN | | Places | | Textures | | Average | |
|---|---|---|---|---|---|---|---|---|---|---|---|
| | | AUROC↑ | FPR95↓ | AUROC↑ | FPR95↓ | AUROC↑ | FPR95↓ | AUROC↑ | FPR95↓ | AUROC↑ | FPR95↓ |
| 336×336 | Neglabel | 99.61 | 1.21 | 96.29 | 22.06 | 93.30 | 32.01 | 89.91 | 40.85 | 94.78 | 24.03 |
| | AdaNeg+Neglabel | 99.81 | 0.46 | 98.35 | 6.66 | 93.15 | 33.03 | 96.17 | 27.61 | 96.84 | 16.94 |
| | Ours | 99.94 | 0.22 | 98.99 | 4.02 | 96.85 | 24.59 | 98.66 | 16.01 | **98.61** | **11.21** |

*Table 7.* Domain-generalizable OOD detection results ImageNet-1K variants as ID.↑ indicates larger values are better and vice versa. The best results in the last two columns are shown in bold.

| ID Dataset | Method | iNaturalist | | SUN | | Places | | Textures | | Average | |
|---|---|---|---|---|---|---|---|---|---|---|---|
| | | AUROC↑ | FPR95↓ | AUROC↑ | FPR95↓ | AUROC↑ | FPR95↓ | AUROC↑ | FPR95↓ | AUROC↑ | FPR95↓ |
| ImageNet-A | Neglabel | 98.80 | 4.09 | 89.83 | 44.38 | 82.88 | 60.10 | 80.25 | 64.34 | 87.94 | 43.23 |
| | AdaNeg+Neglabel | 98.99 | 3.24 | 93.26 | 32.64 | 84.02 | 60.95 | 90.94 | 44.40 | 91.80 | 35.30 |
| | Ours | 99.18 | 3.33 | 97.25 | 13.91 | 92.13 | 40.49 | 96.20 | 26.24 | **96.19** | **20.99** |
| ImageNet-S | Neglabel | 99.34 | 2.24 | 94.93 | 22.73 | 90.78 | 38.62 | 89.29 | 46.10 | 93.59 | 27.42 |
| | AdaNeg+Neglabel | 99.73 | 0.74 | 98.23 | 7.15 | 95.25 | 25.41 | 96.61 | 17.73 | 97.46 | 12.76 |
| | Ours | 99.95 | 0.19 | 99.37 | 5.03 | 96.04 | 23.71 | 98.22 | 9.94 | **98.39** | **10.72** |
| ImageNet-R | Neglabel | 99.58 | 1.60 | 96.03 | 15.77 | 91.97 | 29.48 | 90.60 | 35.67 | 94.54 | 20.63 |
| | AdaNeg+Neglabel | 99.72 | 0.89 | 99.03 | 2.14 | 97.33 | 13.78 | 97.09 | 15.68 | 98.29 | 8.12 |
| | Ours | 99.73 | 1.12 | 99.32 | 2.31 | 97.65 | 10.61 | 98.70 | 5.59 | **98.85** | **4.91** |

*Table 8.* OOD detection results on ImageNet-1k with prompts learned by LAPT (Zhang et al., 2025). Following AdaNeg (Zhang & Zhang, 2024), the performance is measured by FPR95↓. The best results are shown in bold.

| Methods | iNaturalist | SUN | Places | Textures | Average |
|---|---|---|---|---|---|
| NegLabel | 1.10 | 20.59 | 35.38 | 40.11 | 24.29 |
| AdaNeg+NegLabel | 0.58 | 9.98 | 30.47 | 25.25 | 16.32 |
| Ours | 0.31 | 4.41 | 22.32 | 13.60 | **10.16** |

optimization problem while SAL (Du et al., 2024) separates candidate outliers from the unlabeled wild data and then trains a binary classifier using the candidate outliers and the labelled ID data.

## 6.2. CLIP-based Out-of-distribution Detection

The core of CLIP-based OOD detection lies in how to leverage textual supervision with pre-trained VLMs to assist OOD detection on the visual domain. On the one hand, the pioneering work, MCM (Ming et al., 2022), defines textual features as concept proto-types for each ID class and uses the scaled distance between visual features and the closest ID prototype to measure OOD uncertainty. Instead of relying on textual information from only ID label space, ZOC (Esmaeilpour et al., 2022) applies VLMs to discern OOD instances by training a captioner that generates potential OOD labels. Nevertheless, this captioner often fails to produce effective OOD labels, particularly for ID datasets containing many classes. Differently, NegLabel (Jiang et al., 2024) incorporates additional negative class names mined from available data sources as negative proxies. Considering the nonalignment between target visual OOD distri-

bution and the generated negative textual OOD distribution, AdaNeg (Zhang & Zhang, 2024) leverages the benefits of test-time adaptation to generate adaptive proxies by exploring potential OOD images during testing. More recently, Peng et al. (2026b) understand CLIP-based post-hoc OOD detection from an information-theoretical perspective while Peng et al. (2026c) improve negative mining strategy in CLIP-based post-hoc OOD detection from a positive-unlabeled perspective. On the other hand, CLIP-based OOD detection can also be improved by prompt representation learning. In particular, LoCoOp (Miyai et al., 2024) learns ID text prompts by pushing them away from the portions of CLIP local features that have ID-irrelevant nuisances (e.g., backgrounds). CLIPN (Wang et al., 2023a) and LSN (Nie et al., 2024) design a learnable "no" prompt and a "no" text encoder to capture negation semantics within images. Differently, LAPT (Zhang et al., 2025) initializes prompts with negative labels, followed by tuning prompts with cross-modal and cross-distribution mixing.

## 7. Conclusion

This paper theoretically shows that a key assumption behind zero-shot, post-hoc OOD detection with pre-trained vision–language models—that text embeddings of class names can serve as reliable visual class prototypes—is generally invalid due to an intrinsic modality gap between textual prototypes and optimal visual prototypes in the image feature space. Building on this theoretical insight, we propose an online pseudo-supervised framework that respects the post-hoc constraint by learning and updating prototypes directly in the visual feature space from unlabeled test-time

streams and soft CLIP predictions, with accompanying convergence guarantees. Extensive experiments across standard benchmarks demonstrate that this modality-aware prototype calibration consistently improves OOD detection performance and achieves state-of-the-art results.

## Impact Statement

This paper presents work aimed at advancing the field of machine learning. While our research may have a range of potential societal implications, we do not believe that any require specific discussion at this time.

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

# A. More Related Work

Pretraining on large-scale image–text pairs has established vision–language models (VLMs) as a standard backbone for multimodal transfer. Architecturally, existing VLMs can broadly be grouped into two categories: (i) *single-stream* models, which feed concatenated visual and textual features into a unified transformer, such as VisualBERT (Li et al., 2019) and ViLT (Kim et al., 2021); and (ii) *dual-stream* models, which maintain separate visual and textual encoders and learn cross-modal alignment through contrastive image–text pairing, such as CLIP (Radford et al., 2021), ALIGN (Jia et al., 2021), SigLIP (Zhai et al., 2023), and FILIP (Yao et al., 2021). Among these, CLIP-based models have been widely adopted and have inspired numerous follow-up studies aimed at improving data efficiency and downstream adaptation (Yu et al., 2026c; 2020; 2025; 2026a;b; Liao et al., 2026a; Peng et al., 2025a; 2026a). In this paper, we adopt CLIP as the pretrained backbone; however, our method is generally applicable to contrastive vision–language models that learn aligned visual and textual representations.

# B. Proof of Theorem 1

*Proof.* **Step 1: Euclidean gradient.** Rewrite the loss:

$$\mathcal{L}(\mathbf{W}) = -\sum_{i=1}^{N}\left(\frac{1}{\kappa}\mathbf{z}_i \cdot \mathbf{w}_{\hat{y}_i} - \log\sum_{\hat{y}\in\hat{Y}}\exp(\mathbf{z}_i \cdot \mathbf{w}_{\hat{y}}/\kappa)\right). \tag{16}$$

Differentiating $\mathcal{L}(\mathbf{W})$ with respect to $\mathbf{w}_{\hat{y}}$ yields:

$$\nabla_{\mathbf{w}_{\hat{y}}}\mathcal{L}(\mathbf{W}) = \frac{1}{\kappa}\left(\underbrace{\sum_{(\mathbf{x}_i,\hat{y}_i)\in D_{\hat{y}}}(\pi_{i\hat{y}}-1)\mathbf{z}_i + \sum_{(\mathbf{x}_i,\hat{y}_i)\notin D_{\hat{y}}}\pi_{i\hat{y}}\mathbf{z}_i}_{-v_{\hat{y}}(\mathbf{W})}\right), \tag{17}$$

where

$$\pi_{i\hat{y}} = \frac{\exp(\mathbf{z}_i \cdot \mathbf{w}_{\hat{y}}/\kappa)}{\sum_{y\in\hat{Y}}\exp(\mathbf{z}_i \cdot \mathbf{w}_y/\kappa)}. \tag{18}$$

**Step 2: Lagrangian and KKT stationarity on the sphere.** Introduce Lagrange multipliers $\mathbf{u} = \{\mu_{\hat{y}}\}_{\hat{y}\in\hat{\mathcal{y}}}$ for the equality constraints $\|\mathbf{w}_{\hat{y}}\|_2^2 = 1$ and form the Lagrangian

$$\mathcal{L}(\mathbf{W},\mathbf{u}) \triangleq \mathcal{L}(\mathbf{W}) + \sum_{\hat{y}\in\hat{\mathcal{y}}}\mu_{\hat{y}}\big(\|\mathbf{w}_{\hat{y}}\|_2^2 - 1\big). \tag{19}$$

At a constrained optimum $(\mathbf{W}^*, \mathbf{u}^*)$, KKT stationarity requires, for each $\hat{y}\in\hat{\mathcal{y}}$,

$$\nabla_{\mathbf{w}_{\hat{y}}}\mathcal{L}(\mathbf{W}^*,\mathbf{u}^*) = \nabla_{\mathbf{w}_{\hat{y}}}\mathcal{L}(\mathbf{W}) + 2\mu_{\hat{y}}^*\mathbf{w}_{\hat{y}}^* = -\frac{1}{\kappa}v_{\hat{y}}(W^*) + 2\mu_{\hat{y}}^*\mathbf{w}_{\hat{y}}^* = 0, \tag{20}$$

such that

$$-\frac{1}{\kappa}v_{\hat{y}}(\mathbf{W}^*) + 2\mu_{\hat{y}}^*\mathbf{w}_{\hat{y}}^* = 0 \quad\Longrightarrow\quad v_{\hat{y}}(\mathbf{W}^*) = 2\kappa\mu_{\hat{y}}^*\mathbf{w}_{\hat{y}}^*. \tag{21}$$

Hence, $w_{\hat{y}}^*$ must be *parallel* to $v_{\hat{y}}(W^*)$.

**Step 3: Enforce unit norm and obtain the closed form.** If $v_{\hat{y}}(\mathbf{W}^*) \neq 0$, then (21) implies that there exists a scalar $\lambda_{\hat{y}} \neq 0$ such that

$$\mathbf{w}_{\hat{y}}^* = \lambda_{\hat{y}}\, v_{\hat{y}}(\mathbf{W}^*). \tag{22}$$

Imposing $\|\mathbf{w}_{\hat{y}}^*\|_2 = 1$ gives

$$1 = \|\mathbf{w}_{\hat{y}}^*\|_2 = |\lambda_{\hat{y}}|\,\|v_{\hat{y}}(\mathbf{W}^*)\|_2 \quad\Longrightarrow\quad |\lambda_{\hat{y}}| = \frac{1}{\|v_{\hat{y}}(\mathbf{W}^*)\|_2}. \tag{23}$$

Absorbing the sign into $\mu_{\hat{y}}^*$ (equivalently choosing the representative on the ray), we obtain

$$\mathbf{w}_{\hat{y}}^* = \frac{v_{\hat{y}}(\mathbf{W}^*)}{\|v_{\hat{y}}(\mathbf{W}^*)\|_2} = \ell_2 \left( \sum_{i \in D_{\hat{y}}} (1 - \pi_{i\hat{y}})\mathbf{z}_i - \sum_{i \notin D_{\hat{y}}} \pi_{i\hat{y}}\mathbf{z}_i \right), \tag{24}$$

which is precisely the statement of Theorem 1.

If $v_{\hat{y}}(\mathbf{W}^*) = 0$, then (20) holds with $\mu_{\hat{y}}^* = 0$ for any unit vector $\mathbf{w}_{\hat{y}}^*$, and the normalization in (24) is undefined. Thus, the closed form (24) applies for each $\hat{y}$ such that $v_{\hat{y}}(\mathbf{W}^*) \neq 0$. $\quad\square$

## C. Proof of Theorem 2

*Proof.* Fix any label $\hat{y} \in \widehat{\mathcal{Y}}$. For simplicity, let us denote

$$\mathbf{r} := \mathbf{r}_{\hat{y}}, \qquad \mathbf{w}^* := \mathbf{w}_{\hat{y}}^\star. \tag{25}$$

Since both textual and visual prototypes lie on a unit hyper-sphere $\mathbb{S}^{d-1}$, we have

$$\|\mathbf{r}\|_2 = 1, \qquad \|\mathbf{w}^*\|_2 = 1. \tag{26}$$

Moreover, Theorem 1 implies $\mathbf{w} \in S = \operatorname{span}\{\mathbf{z}_1, \ldots, \mathbf{z}_N\}$. By orthogonal decomposition, we can write

$$\mathbf{r} = \operatorname{Proj}_{\mathcal{S}}(\mathbf{r}) + \operatorname{Proj}_{\mathcal{S}^\perp}(\mathbf{r}). \tag{27}$$

For simplicity, let us denote

$$\mathbf{r}_{\mathcal{S}} := \operatorname{Proj}_{\mathcal{S}}(\mathbf{r}), \qquad \mathbf{r}_{\mathcal{S}^\perp} := \operatorname{Proj}_{\mathcal{S}^\perp}(\mathbf{r}). \tag{28}$$

Then

$$\mathbf{r} = \mathbf{r}_{\mathcal{S}} + \mathbf{r}_{\mathcal{S}^\perp}, \qquad \mathbf{r}_{\mathcal{S}} \perp \mathbf{r}_{\mathcal{S}^\perp}. \tag{29}$$

Since $\mathbf{w}^* \in \mathcal{S}$, we have $\mathbf{r}_{\mathcal{S}} - \mathbf{w}^* \in \mathcal{S}$. Therefore,

$$\mathbf{r}_{\mathcal{S}} - \mathbf{w}^* \perp \mathbf{r}_{\mathcal{S}^\perp}. \tag{30}$$

It follows from the Pythagorean theorem that

$$\begin{aligned}
\|\mathbf{r} - \mathbf{w}^*\|_2^2 &= \|\mathbf{r}_{\mathcal{S}} + \mathbf{r}_{\mathcal{S}^\perp} - \mathbf{w}^*\|_2^2 \\
&= \|(\mathbf{r}_{\mathcal{S}} - \mathbf{w}^*) + \mathbf{r}_{\mathcal{S}^\perp}\|_2^2 \\
&= \|\mathbf{r}_{\mathcal{S}} - \mathbf{w}^*\|_2^2 + \|\mathbf{r}_{\mathcal{S}^\perp}\|_2^2.
\end{aligned} \tag{31}$$

Expanding the first term gives

$$\|\mathbf{r} - \mathbf{w}^*\|_2^2 = \|\mathbf{r}_{\mathcal{S}}\|_2^2 + \|\mathbf{w}^*\|_2^2 - 2\mathbf{r}_{\mathcal{S}}^\top \mathbf{w}^* + \|\mathbf{r}_{\mathcal{S}^\perp}\|_2^2. \tag{32}$$

Using $\|\mathbf{r}_{\mathcal{S}}\|_2^2 + \|\mathbf{r}_{\mathcal{S}^\perp}\|_2^2 = \|\mathbf{r}\|_2^2 = 1$ and $\|\mathbf{w}^*\|_2^2 = 1$, we obtain

$$\|\mathbf{r} - \mathbf{w}^*\|_2^2 = 2 - 2\mathbf{r}_{\mathcal{S}}^\top \mathbf{w}^*. \tag{33}$$

By the Cauchy–Schwarz inequality,

$$\mathbf{r}_{\mathcal{S}}^\top \mathbf{w}^* \leq \|\mathbf{r}_{\mathcal{S}}\|_2 \|\mathbf{w}^*\|_2 = \|\mathbf{r}_{\mathcal{S}}\|_2. \tag{34}$$

Therefore,

$$\|\mathbf{r} - \mathbf{w}^*\|_2^2 \geq 2 - 2\|\mathbf{r}_{\mathcal{S}}\|_2. \tag{35}$$

Returning to the original notation, we have

$$\|\mathbf{r}_{\hat{y}} - \mathbf{w}_{\hat{y}}^\star\|_2^2 \geq 2\left(1 - \|\operatorname{Proj}_{\mathcal{S}}(\mathbf{r}_{\hat{y}})\|_2\right). \tag{36}$$

Summing over all $\hat{y} \in \widehat{\mathcal{Y}}$ yields

$$\|\mathbf{R} - \mathbf{W}^{\star}\|_F^2 = \sum_{\hat{y} \in \widehat{\mathcal{Y}}} \|\mathbf{r}_{\hat{y}} - \mathbf{w}_{\hat{y}}^{\star}\|_2^2 \geq \sum_{\hat{y} \in \widehat{\mathcal{Y}}} 2\left(1 - \|\mathrm{Proj}_{\mathcal{S}}(\mathbf{r}_{\hat{y}})\|_2\right). \tag{37}$$

Finally, since $\mathbf{r}_{\hat{y}} \in \mathbb{S}^{d-1}$, we have

$$\|\mathrm{Proj}_{\mathcal{S}}(\mathbf{r}_{\hat{y}})\|_2^2 + \mathrm{Proj}_{\mathcal{S}^{\perp}}(\mathbf{r}_{\hat{y}})\|_2^2 = \|\mathbf{r}_{\hat{y}}\|_2^2 = 1. \tag{38}$$

Hence,

$$\|\mathrm{Proj}_{\mathcal{S}}(\mathbf{r}_{\hat{y}})\|_2 = \sqrt{1 - \|\mathrm{Proj}_{\mathcal{S}^{\perp}}(\mathbf{r}_{\hat{y}})\|_2^2}. \tag{39}$$

Substituting this identity into the previous bound gives

$$\|\mathbf{R} - \mathbf{W}^{\star}\|_F^2 \geq \sum_{\hat{y} \in \widehat{\mathcal{Y}}} 2\left(1 - \sqrt{1 - \|\mathrm{Proj}_{\mathcal{S}^{\perp}}(\mathbf{r}_{\hat{y}})\|_2^2}\right). \tag{40}$$

$\square$

# D. Proof of Theorem 3

*Proof.* **Step 1: One-step progress inequality.** The update in Eq. (8) can be rewritten as a projected gradient descent, i.e.,

$$\mathbf{W}^{(i)} = \Pi_{\mathcal{K}}\left(\mathbf{W}^{(i-1)} - \eta_i \mathbf{G}^{(i)}\right), \tag{41}$$

where $\Pi_{\mathcal{K}}$ is Euclidean projection onto the feasible set $\mathcal{K} \triangleq (\mathbb{S}^{d-1})^M$ and $\mathbf{G}^{(i)} \triangleq \nabla_{\mathbf{W}} \hat{\ell}(\mathbf{W}^{(i-1)}; \mathbf{x}_i, \hat{y})$

Projection is non-expansive, so for any $\mathbf{W} \in \mathcal{K}$,

$$\|\mathbf{W}^{(i)} - \mathbf{W}\|_F^2 = \left\|\Pi_{\mathcal{K}}(\mathbf{W}^{(i-1)} - \eta_i \mathbf{G}^{(i)}) - \Pi_{\mathcal{K}}(\mathbf{W})\right\|_F^2 \tag{42}$$

$$\leq \|\mathbf{W}^{(i-1)} - \eta_i \mathbf{G}^{(i)} - \mathbf{W}\|_F^2 \tag{43}$$

$$= \|\mathbf{W}^{(i-1)} - \mathbf{W}\|_F^2 - 2\eta_i \langle \mathbf{G}^{(i)}, \mathbf{W}^{(i-1)} - \mathbf{W} \rangle_F + \eta_i^2 \|\mathbf{G}^{(i)}\|_F^2. \tag{44}$$

Rearranging gives

$$\langle \mathbf{G}^{(i)}, \mathbf{W}^{(i-1)} - \mathbf{W} \rangle_F \leq \frac{\|\mathbf{W}^{(i-1)} - \mathbf{W}\|_F^2 - \|\mathbf{W}^{(i)} - \mathbf{W}\|_F^2}{2\eta_i} + \frac{\eta_i}{2}\|\mathbf{G}^{(i)}\|_F^2. \tag{45}$$

**Step 2: Use convexity to relate inner products to regret.** By convexity of $\hat{\ell}(\mathbf{W}; \mathbf{x}_i, \hat{y})$ w.r.t. $\mathbf{W}$,

$$\hat{\ell}(\mathbf{W}^{(i-1)}; \mathbf{x}_i, \hat{y}) - \hat{\ell}(\mathbf{W}^*; \mathbf{x}_i, \hat{y}) \leq \langle \mathbf{G}^{(i)}, \mathbf{W}^{(i-1)} - \mathbf{W}^* \rangle_F. \tag{46}$$

Combine with the inequality above and sum over $i = 1$ to $N$:

$$\sum_{i=1}^{N} \left(\hat{\ell}(\mathbf{W}^{(i-1)}; \mathbf{x}_i, \hat{y}) - \hat{\ell}(\mathbf{W}^*; \mathbf{x}_i, \hat{y})\right) \leq \sum_{i=1}^{N} \frac{\|\mathbf{W}^{(i-1)} - \mathbf{W}^*\|_F^2 - \|\mathbf{W}^{(i)} - \mathbf{W}^*\|_F^2}{2\eta_i} + \sum_{i=1}^{N} \frac{\eta_i}{2}\|\mathbf{G}^{(i)}\|_F^2. \tag{47}$$

**Step 3: Bound the diameter $D$ of $\mathcal{K}$.** Let $D \triangleq \sup_{\mathbf{U}, \mathbf{V} \in \mathcal{K}} \|\mathbf{U} - \mathbf{V}\|_F$. For any $\mathbf{U} = [\mathbf{u}_1, \dots, \mathbf{u}_M]$, $\mathbf{V} = [\mathbf{v}_1, \dots, \mathbf{v}_M] \in \mathcal{K}$,

$$\|\mathbf{U} - \mathbf{V}\|_F^2 = \sum_{j=1}^{M} \|\mathbf{u}_j - \mathbf{v}_j\|_2^2. \tag{48}$$

Since $\|\mathbf{u}_j\|_2 = \|\mathbf{v}_j\|_2 = 1$, we have $\|\mathbf{u}_j - \mathbf{v}_j\|_2 \leq 2$ for each $j$, hence

$$\|\mathbf{U} - \mathbf{V}\|_F^2 \leq \sum_{j=1}^{M} 4 = 4M \quad \Rightarrow \quad D \leq 2\sqrt{M}. \tag{49}$$

In particular, $\|\mathbf{W}^{(0)} - \mathbf{W}\|_F \leq D$ for any $\mathbf{W} \in \mathcal{K}$.

**Step 4: Bound the two sums.** Since $\eta_i = \rho/\sqrt{i}$ is non-increasing, the telescoping term satisfies

$$\sum_{i=1}^{N} \frac{\|\mathbf{W}^{(i-1)} - \mathbf{W}^*\|_F^2 - \|\mathbf{W}^{(i)} - \mathbf{W}^*\|_F^2}{2\eta_i} \leq \frac{\|\mathbf{W}^{(0)} - \mathbf{W}^*\|_F^2}{2\eta_N} \leq \frac{D^2}{2\eta_N}. \tag{50}$$

For the second term, use $\|\mathbf{G}^{(i)}\|_F \leq \epsilon$:

$$\sum_{i=1}^{N} \frac{\eta_i}{2} \|\mathbf{G}^{(i)}\|_F^2 \leq \frac{\epsilon^2}{2} \sum_{i=1}^{N} \eta_i = \frac{\epsilon^2 \rho}{2} \sum_{i=1}^{N} \frac{1}{\sqrt{i}} \leq \frac{\epsilon^2 \rho}{2} \cdot 2\sqrt{N} = \epsilon^2 \rho \sqrt{N}. \tag{51}$$

Also, $\eta_N = \rho/\sqrt{N}$, hence

$$\frac{D^2}{2\eta_N} = \frac{D^2}{2} \cdot \frac{\sqrt{N}}{\rho} \leq 2M \frac{\sqrt{N}}{\rho}. \tag{52}$$

**Step 5: Conclude regret bound.** Putting everything together yields

$$\sum_{i=1}^{N} \left( \hat{\ell}(\mathbf{W}^{(i-1)}; \mathbf{x}_i, \hat{\mathcal{Y}}) - \hat{\ell}(\mathbf{W}^*; \mathbf{x}_i, \hat{\mathcal{Y}}) \right) \leq \underbrace{\left( \frac{2M}{\rho} + \epsilon^2 \rho \right) \sqrt{N}}_{O(\sqrt{N})}. \tag{53}$$

$\square$

# E. Online Prototype Learning with ID-only labels for OOD Detection

To realize the idea, we need to first identify positive samples, i.e., those with high confidence of being ID, from the test data stream. To this end, we adopt a hard thresholding rule based on the off-the-shelf $S_{\text{MCM}}(\mathbf{x}_i; f)$ in Eq. (2), i.e.,

$$\text{Positive} : S_{\text{MCM}}(\mathbf{x}_i; f) \geq \beta, \tag{54}$$

where $\beta \in [0, 1]$ is a hyper-parameter.

Given the $i$-th test sample $\mathbf{x}_i$ ($i = 1, 2, \ldots$) to arrive, we use $c_i^+$ to represent the cumulative number of positive samples we have ever seen so far, such that

$$c_i^+ = c_{i-1}^+ + \mathbb{1}\left( S_{\text{MCM}}(\mathbf{x}_i; f) \geq \beta \right), \tag{55}$$

with initialization $c_0^+ = 0$ and $\mathbb{1}(\cdot)$ as an indicator function.

For each $y \in \mathcal{Y}_{\text{I}}$, the class prototype $\mathbf{w}_y$ is updated as

$$\mathbf{w}_y^{(i)} = \begin{cases} \mathbf{w}_y^{(i-1)} & \text{if } S_{\text{MCM}}(\mathbf{x}_i; f) < \beta, \\ \ell_2 \left( \mathbf{w}_y^{(i-1)} - \eta_i^+ \frac{\partial \hat{\ell}(\mathbf{W}^{(i-1)}; \mathbf{x}_i, \mathcal{Y}_{\text{I}})}{\partial \mathbf{w}_y^{(i-1)}} \right) & \text{otherwise,} \end{cases} \tag{56}$$

where $\eta_i^+ = \rho/\sqrt{c_i^+}$.

Finally, we can arrive at the online OOD scoring function for $\mathbf{x}_i$ as follows:

$$S_{\text{ours}}(\mathbf{x}_i; f) = \frac{\max_{y \in \mathcal{Y}_{\text{I}}} \exp(\mathbf{z}_i \cdot \mathbf{w}_y^{(i)}/\tau)}{\sum_{y \in \mathcal{Y}_{\text{I}}} \exp(\mathbf{z}_i \cdot \mathbf{w}_y^{(i)}/\tau)}, \tag{57}$$

with initialization satisfies $\mathbf{w}_y^{(0)} = f_{\mathcal{T}}(\mathcal{P}(y))$ for each $y \in \mathcal{Y}_{\text{I}}$. For clarity, we summarize the complete algorithm in Algorithm 2. Empirical comparison can be found in Table 9.

---

**Algorithm 2** Online Prototype Learning with ID-only labels for OOD Detection

---

1: **Input:** Pre-trained CLIP-based model $f$, Prompt template $\mathcal{P}(\cdot)$, ID labels $\mathcal{Y}_{\mathrm{I}} = \{y_1, \dots, y_K\}$.
2: Initialize $c_0^+ = 0$
3: Initialize $\mathbf{w}_y^{(0)} = f_{\mathcal{T}}\big(\mathcal{P}(y)\big)$ for each $y \in \mathcal{Y}_{\mathrm{I}}$
4: **while** the $i$-th test sample $\mathbf{x}_i$ arrives **do**
5:     Update $c_i^+$ via Eq. (55)
6:     Update $\mathbf{w}_y^{(i)}$ via Eq. (56), $\forall y \in \mathcal{Y}_{\mathrm{I}}$
7:     Compute $S_{\mathrm{ours}}(\mathbf{x}_i; f)$ via Eq. (57)
8: **end while**

---

*Table 9.* OOD detection results on ImageNet-1K, where a VIT B/16 CLIP encoder is adopted. $\uparrow$ indicates larger values are better and vice versa. The best results in the last two columns are shown in bold.

| Dataset | iNaturalist | | SUN | | Places | | Textures | | Average | |
|---|---|---|---|---|---|---|---|---|---|---|
| Metric | AUROC↑ | FPR95↓ | AUROC↑ | FPR95↓ | AUROC↑ | FPR95↓ | AUROC↑ | FPR95↓ | AUROC↑ | FPR95↓ |
| **Zero-Shot Training-free Methods with Only ID Labels** | | | | | | | | | | |
| Mahalanobis | 55.89 | 99.33 | 59.94 | 99.41 | 65.96 | 98.54 | 64.23 | 98.46 | 61.50 | 98.94 |
| Energy | 85.09 | 81.08 | 84.24 | 79.02 | 83.38 | 75.08 | 65.56 | 93.65 | 79.57 | 82.21 |
| ZOC | 86.09 | 87.30 | 81.20 | 81.51 | 83.39 | 73.06 | 76.46 | 98.90 | 81.79 | 85.19 |
| MCM | 94.59 | 32.20 | 92.25 | 38.80 | 90.31 | 46.20 | 86.12 | 58.50 | 90.82 | 43.93 |
| GL-MCM | 96.71 | 15.16 | 93.41 | 29.16 | 90.37 | 37.07 | 83.11 | 55.85 | 90.90 | 35.06 |
| Ours (Eq. (57)) | 96.18 | 7.45 | 93.45 | 27.80 | 88.61 | 44.17 | 90.27 | 42.50 | **92.13** | **30.48** |

# F. Evaluation Metrics.

We evaluate the performance of OOD detection with two widely used metrics, including 1) the false positive rate of OOD data is measured when the true positive rate of ID data reaches 95% (FPR95); 2) the area under the receiver operating curve (AUROC) is computed to quantify the probability of the ID case receiving a higher score than OOD case.

# G. More Ablation Studies

**Temporal Shift.** Since our method dynamically updates class prototypes in a test-time adaptation manner, it is crucial to investigate its stability under temporal shifts, where OOD environments evolve over time. Specifically, we use ImageNet as the ID dataset and assume that OOD datasets change sequentially over time (e.g., I–S–P–T represents OOD dataset transitions from iNaturalist $\rightarrow$ SUN $\rightarrow$ Places $\rightarrow$ Textures). In implementation, we only initialize class prototypes via Eq. (15) once at the very beginning rather than when processing a new OOD dataset. As shown in Table 10, our method consistently maintains a large advantage over the AdaNeg+NegLabel baseline, validating its robustness to temporal shifts.

**Sample Order.** In our experiments, ID and OOD samples are randomly shuffled during testing, corresponding to the "Random Shuffled" scenario. To analyze the impact of sample order, we conducted additional experiments. Specifically, we consider two extreme cases: "ID First" (all ID samples are tested before any OOD samples) and "OOD First" (all OOD samples are tested before any ID samples). As shown in Table 11, while performance does drop under these extreme settings compared to the "Random Shuffled" scenario, our method still significantly outperforms the AdaNeg+NegLabel baseline. This demonstrates the robustness and effectiveness of our method, even when the order of test samples is highly imbalanced.

# H. Analysis on Memory and Time Complexity

We analyze the memory and time complexity of the proposed method and compare it with relevant baselines in Table 12 to clarify its computational overhead and scalability. Note that we omit the computation cost introduced by extracting features from pre-trained CLIP, as our method keeps the same feature extraction procedure as AdaNeg (Zhang & Zhang, 2024).

*Table 10.* OOD detection performance under temporal shifts on ImageNet-1K.

| Methods | iNaturalist | | SUN | | Places | | Textures | | Average | |
|---|---|---|---|---|---|---|---|---|---|---|
| | AUROC↑ | FPR95↓ | AUROC↑ | FPR95↓ | AUROC↑ | FPR95↓ | AUROC↑ | FPR95↓ | AUROC↑ | FPR95↓ |
| AdaNeg+NegLabel | 99.71 | 0.59 | 97.44 | 9.50 | 94.55 | 34.34 | 94.93 | 31.27 | 96.66 | 18.92 |
| I-S-P-T | 99.69 | 0.72 | 98.35 | 5.76 | 94.69 | 29.01 | 97.00 | 17.99 | 97.43 | 13.37 |
| S-P-T-I | 99.81 | 0.52 | 98.79 | 5.03 | 95.19 | 28.66 | 97.46 | 16.82 | 97.81 | 12.76 |
| P-T-I-S | 99.77 | 0.55 | 98.83 | 5.89 | 94.64 | 29.61 | 96.76 | 17.93 | 97.50 | 13.50 |
| T-I-S-P | 99.83 | 0.51 | 98.95 | 4.94 | 95.51 | 27.53 | 97.13 | 17.40 | 97.86 | 12.60 |

*Table 11.* OOD detection results on ImageNet-1k with different sample order. ↑ indicates larger values are better and vice versa.

| Methods | iNaturalist | | SUN | | Places | | Textures | | Average | |
|---|---|---|---|---|---|---|---|---|---|---|
| | AUROC↑ | FPR95↓ | AUROC↑ | FPR95↓ | AUROC↑ | FPR95↓ | AUROC↑ | FPR95↓ | AUROC↑ | FPR95↓ |
| AdaNeg+NegLabel | 99.71 | 0.59 | 97.44 | 9.50 | 94.55 | 34.34 | 94.93 | 31.27 | 96.66 | 18.92 |
| ID First | 99.90 | 0.52 | 98.91 | 5.12 | 95.07 | 29.13 | 96.96 | 18.35 | 97.71 | 13.28 |
| OOD First | 99.89 | 0.52 | 98.85 | 5.39 | 94.86 | 30.75 | 96.90 | 18.35 | 97.63 | 13.75 |
| Random Shuffle | 99.88 | 0.50 | 98.94 | 5.00 | 95.17 | 28.55 | 97.01 | 17.12 | 97.75 | 12.79 |

*Table 12.* Analysis on the memory and time complexity, where computational cost is measured in seconds on an NVIDIA A800 GPU.

| Method | Memory Complexity | Computation Cost | | | | |
|---|---|---|---|---|---|---|
| | | iNaturalist | SUN | Places | Textures | Total |
| AdaNeg+NegLabel | $2d(K+L)$ | 38.05 | 37.38 | 52.75 | 33.87 | 193.02 |
| Ours (Eq. (14)) | $d(K+L)$ | 23.20 | 23.13 | 40.17 | 20.27 | 107.06 |

*Table 13.* Mean and standard deviation of OOD detection performance across various random seeds using CLIP-B/16 on ImageNet-1k as the ID dataset. ↑ indicates larger values are better and vice versa.

| Seeds | iNaturalist | | SUN | | Places | | Textures | | Average | |
|---|---|---|---|---|---|---|---|---|---|---|
| | AUROC↑ | FPR95↓ | AUROC↑ | FPR95↓ | AUROC↑ | FPR95↓ | AUROC↑ | FPR95↓ | AUROC↑ | FPR95↓ |
| 0 | 99.89 | 0.52 | 98.94 | 5.00 | 95.18 | 28.63 | 97.08 | 17.57 | 97.77 | 12.93 |
| 1 | 99.89 | 0.49 | 98.94 | 4.79 | 95.09 | 28.91 | 97.16 | 16.40 | 97.77 | 12.65 |
| 2 | 99.77 | 0.56 | 98.93 | 5.19 | 95.13 | 28.76 | 97.06 | 17.16 | 97.72 | 12.92 |
| 3 | 99.89 | 0.51 | 98.90 | 4.92 | 95.06 | 28.84 | 97.00 | 18.82 | 97.71 | 13.27 |
| 4 | 99.89 | 0.49 | 98.91 | 4.83 | 95.09 | 28.64 | 97.06 | 17.34 | 97.74 | 12.83 |
| 5 | 99.90 | 0.53 | 98.90 | 5.05 | 95.01 | 29.22 | 96.96 | 17.20 | 97.69 | 13.00 |
| 6 | 99.91 | 0.45 | 99.01 | 5.64 | 95.46 | 27.93 | 96.28 | 17.16 | 97.65 | 12.75 |
| 7 | 99.89 | 0.50 | 98.90 | 5.03 | 95.10 | 28.70 | 97.07 | 16.86 | 97.74 | 12.77 |
| 8 | 99.89 | 0.49 | 98.93 | 4.89 | 95.09 | 28.74 | 97.10 | 16.68 | 97.76 | 12.70 |
| 9 | 99.91 | 0.45 | 99.01 | 4.64 | 95.46 | 27.13 | 97.28 | 15.96 | 97.91 | 12.05 |
| Mean | 99.88 | 0.50 | 98.94 | 5.00 | 95.17 | 28.55 | 97.01 | 17.12 | 97.75 | 12.79 |
| Std | 0.04 | 0.03 | 0.04 | 0.27 | 0.16 | 0.59 | 0.27 | 0.77 | 0.07 | 0.31 |

*Table 14.* Detailed OOD detection results on ImageNet-1k. ↑ indicates larger values are better and vice versa. The best results are in bold.

| Near-/Far-OOD | Datasets | AdaNeg+NegLabel | | Ours | |
|---|---|---|---|---|---|
| | | AUROC↑ | FPR95↓ | AUROC↑ | FPR95↓ |
| Near-OOD | SSB-hard | 75.11 | 74.91 | 83.64 | 55.11 |
| | NINCO | 78.30 | 60.10 | 87.96 | 50.60 |
| | Average | 76.70 | 67.51 | **85.80** | **52.86** |
| Far-OOD | iNaturalist | 99.72 | 0.72 | 99.81 | 0.57 |
| | Textures | 95.71 | 21.40 | 97.80 | 15.17 |
| | OpenImage-O | 93.87 | 29.81 | 95.40 | 23.91 |
| | Average | 96.43 | 17.31 | **97.67** | **13.22** |

*Table 15.* Detailed OOD detection results on CIFAR-10. ↑ indicates larger values are better and vice versa. The best results are in bold.

| Near-/Far-OOD | Datasets | AdaNeg+NegLabel | | Ours | |
|---|---|---|---|---|---|
| | | AUROC↑ | FPR95↓ | AUROC↑ | FPR95↓ |
| Near-OOD | CIFAR-100 | 90.93 | 35.80 | 92.66 | 29.23 |
| | TIN | 98.63 | 5.01 | 98.77 | 3.53 |
| | Average | 94.78 | 20.40 | **95.72** | **16.38** |
| Far-OOD | MNIST | 99.96 | 0.13 | 99.98 | 0.02 |
| | SVHN | 99.87 | 0.04 | 99.88 | 0.04 |
| | Texture | 99.82 | 0.04 | 99.90 | 0.02 |
| | Places365 | 97.29 | 10.93 | 99.56 | 8.92 |
| | Average | 99.26 | 2.79 | **99.83** | **2.25** |

*Table 16.* Detailed OOD detection results on CIFAR-100. ↑ indicates larger values are better and vice versa. The best results are in bold.

| Near-/Far-OOD | Datasets | AdaNeg+NegLabel | | Ours | |
|---|---|---|---|---|---|
| | | AUROC↑ | FPR95↓ | AUROC↑ | FPR95↓ |
| Near-OOD | CIFAR-10 | 79.91 | 58.24 | 85.63 | 42.18 |
| | TIN | 89.34 | 59.90 | 93.09 | 49.23 |
| | Average | 84.62 | 59.07 | **89.36** | **45.70** |
| Far-OOD | MNIST | 97.90 | 4.18 | 99.12 | 2.30 |
| | SVHN | 97.60 | 6.03 | 98.98 | 3.78 |
| | Texture | 95.14 | 30.00 | 97.20 | 16.88 |
| | Places365 | 90.35 | 77.20 | 92.31 | 64.93 |
| | Average | 95.25 | 29.35 | **96.90** | **21.97** |

