# OpenReview forum: "Respecting Modality Gap in Post-hoc Out-of-distribution Detection with Pre-trained Vision-Language Models"
_ICML.cc/2026/Conference — ICML 2026 regular_

### Official Review · Reviewer_dDmN · 2026-02-22

**Soundness:** 3
**Presentation:** 2
**Significance:** 3
**Originality:** 3
**Overall Recommendation:** 4
**Confidence:** 3

**Summary:**

This paper challenges the adopted “text-as-prototype” paradigm by theoretically showing that off-the-shelf textual prototypes are generally misaligned with the optimal visual prototypes, yielding an intrinsic modality gap that cannot be eliminated by prompt engineering alone, and further presents an online pseudo-supervised framework that directly learns class prototypes in the visual feature space using unlabeled test-time data streams and soft predictions from pre-trained VLMs.

**Compliance With Llm Reviewing Policy:**

Affirmed.

**Final Justification:**

After the rebuttal, I will maintain my positive score.

**Key Questions For Authors:**

Please refer to the weakness.

**Limitations:**

Please refer to the weakness.

**Strengths And Weaknesses:**

Strengths:

1. Label-free Optimization: During the testing phase completely without ground-truth labels, it utilizes CLIP's own soft predictions as a "teacher" to teach visual prototypes how to self-calibrate.

2. Respecting the Modality Gap: Instead of forcing text to adapt to vision, it allows prototypes to "leave" the text space and move into the visual subspace.

3. Convergence Guarantee: It proves that the cumulative regret of this online update algorithm is bounded, guaranteeing that as test data increases, the prototypes will become increasingly accurate.

Weaknesses:

The proposed online prototype update mechanism heavily relies on the accuracy of intermediate signals, which raises concerns about potential error accumulation and the stability of the prototype calibration process in the absence of ground-truth labels during test-time streams.

1.  Is there a risk of error accumulation with pseudo-labels, causing the visual prototype update to collapse? If so, what methods are available to mitigate this issue?

2.  If NegLabel itself lacks sufficient capability to distinguish ID/OOD in certain scenarios, will the thresholding decision be incorrect?

---

> ### Author Rebuttal · Authors · 2026-03-27
>
> We thank Reviewer dDmN for valuable comments. As to the questions and suggestions you raise, we took them seriously. Our response is as follows.
>
> # 1. The proposed online prototype update mechanism heavily relies on the accuracy of intermediate signals
>
> While the concern about error accumulation is valid in principle, we argue the concern does not fully apply to our method because the online updates are **selective and theoretically controlled**.
>
> - We do **NOT** update prototypes with every incoming sample. Instead, it employs a highly **conservative** thresholding rule (controlled by $\beta$=0.95 in our method) to filter out ambiguous samples. Only samples with high confidence (likely ID or likely OOD) are selected for updates. Specifically, samples with scores in the range $(1-\beta, \beta)$ are treated as "uncertain" and are excluded from the update process. This design choice prevents noisy or misclassified "intermediate" signals from corrupting the prototypes, thereby significantly mitigating the risk of error accumulation.
> - We provide a formal theoretical foundation to ensure the stability of the online updating in Theorem 4 establishing that the cumulative regret of the online optimization procedure is **upper-bounded** by $O(\sqrt{N})$. This result indicates that the online learner remains controlled relative to the best fixed solution in hindsight, rather than drifting arbitrarily due to early mistakes.
> - The empirical results do **NOT** show the kind of instability one would expect if error accumulation were dominating. The method consistently outperforms prior baselines across ImageNet-1K, OpenOOD, CIFAR, backbone changes, and domain-shifted settings, and the ablation study reports that performance is not overly sensitive to the key hyperparameters.
>
> # 2. Robustness of thresholding decision to capacity of NegLabel
>
> To address your concern, we validate our method under the case where negative labels are words **randomly selected** from WordNet. The table below shows that, in this case, NegLabel suffers from considerable performance drop and therefore has a weaken capacity of thresholding decision. However, even in this extreme case, our method still outperforms AdaNeg+NegLabel.
>
> |FPR95/AUROC|iNaturalist|SUN|Places|Textures|Avg|
> |-|-|-|-|-|-|
> |NegLabel|1.91/99.49|20.53/95.49|35.59/91.64|43.56/90.22|25.40/94.21
> |NegLabel (random selection)|9.11/97.96|28.67/93.93|45.10/89.54|55.87/86.62|34.69/92.01
> |AdaNeg+NegLabel|0.59/99.71|9.50/97.44|34.34/94.55|31.27/94.93|18.92/96.66
> |Ours (random selection)|0.58/99.73|8.91/97.97|30.19/95.83|22.76/95.59|15.61/97.28
> |Ours|0.50/99.88|5.00/98.94|28.55/95.17|17.12/97.01|12.79/97.75

---

> > ### Author Rebuttal · Reviewer_dDmN · 2026-04-01
> >
> > Thanks for the detailed rebuttal.

---

> > > ### Author Response · Authors · 2026-04-01
> > >
> > > Thank you very much for your response and for confirming that all concerns have been addressed. Thank you again for your time and consideration.

---

### Official Review · Reviewer_ABCp · 2026-03-05

**Soundness:** 3
**Presentation:** 3
**Significance:** 2
**Originality:** 3
**Overall Recommendation:** 4
**Confidence:** 3

**Summary:**

This paper studies post-hoc OOD detection with vision-language models (VLMs), focusing on a key limitation of prior “text-as-prototype” approaches. The authors argue that text prototypes are not guaranteed to match the optimal visual prototypes due to a modality gap, and provide theoretical analysis to support this claim. Motivated by this, the paper proposes a training-free, test-time method that updates class prototypes online using pseudo supervision, and extends the label space with additional pseudo-classes for OOD. Experiments on various benchmarks show consistent improvements.

**Compliance With Llm Reviewing Policy:**

Affirmed.

**Final Justification:**

The authors' responses fully address my questions, so I will keep my positive score.

**Key Questions For Authors:**

Please see the Weakness

**Limitations:**

The authors only provide Impact Statements.

**Strengths And Weaknesses:**

Strength

The paper provides a clear formulation of the issue in CLIP-based OOD detection and offers theoretical insights to explain why the text-as-prototype assumption can fail due to a modality gap.

The proposed method is supported by detailed theoretical analysis, which increases confidence in the algorithmic design and stability.

Extensive experiments across multiple benchmarks are provided, demonstrating the effectiveness of the proposed method.

The approach is efficient in practice as it is training-free and only performs lightweight test-time updates on prototypes.

Weakness

The technical novelty is somewhat limited. The core algorithm mainly replaces ground-truth supervision with pseudo labels at test time and performs online prototype updates. In addition, the method strongly relies on an existing OOD scoring function (e.g., NegLabel) to select confident “ID-like” and “OOD-like” samples for updates.

The method is evaluated under standard static benchmark setups. In realistic test-time deployment, the target distribution may drift over time (for example, gradual domain shift). Since the approach performs online updates, its stability and performance under continuously changing target domains should be examined.

---

> ### Author Rebuttal · Authors · 2026-03-27
>
> We appreciate the insightful comments provided by Reviewer ABCp. Please see our responses to your concerns below.
>
> # 1. Technical novelty
>
> We thank the reviewer for the opportunity to clarify our novel contributions.
>
> Our contribution is **NOT** merely the use of pseudo-labels or online updates, but a **critique** of the text-as-prototype paradigm in CLIP-based OOD detection. Overall, a central concept presented by this paper is that textual prototypes are intrinsically misaligned with optimal visual prototypes due to a provable modality gap, challenging a fundamental assumption underlying prior methods. To mitigate this gap, we propose a principled framework that learns modality-consistent visual prototypes under the post-hoc constraint, where pseudo-supervision is derived from a surrogate objective and optimized online with theoretical guarantees.
>
> We emphasize that using off-the-shelf scoring functions for sample selection is indeed **a free lunch** and **follows a standard practice** in TTA-enhanced post-hoc CLIP-based OOD detection (e.g., AdaNeg). In contrast to prior works (e.g., MCM, NegLabel, AdaNeg), which remain within the flawed text-as-prototype paradigm, our method explicitly addresses and replaces this paradigm with modality-consistent prototype learning. The consistent and substantial gains across benchmarks further confirm that correcting the modality gap  drives the improvement.
>
> # 2. Experiments under continuously changing test distribution
> We kindly note that the robustness of our method under temporal shifts is investigated in Table 10 (Appendix E), where the OOD data distribution evolve sequentially over time, e.g., iNaturalist→SUN→Places→Textures (I-S-P-T). In this case, different from AdaNeg, prototypes are initialized only once at the beginning and then updated continuously throughout the stream. For your convenience, we provide the results in the below.
> |FPR95/AUROC|I|S|P|T|Avg|
> |-|-|-|-|-|-|
> |AdaNeg+NegLabel|0.59/99.71|9.50/97.44|34.34/94.55|31.27/94.93|18.92/96.66
> |Ours (I-S-P-T)|0.72/99.69|5.76/98.35|29.01/94.69|17.99/97.00|13.37/97.43
> |Ours (S-P-T-I)|0.52/99.87|5.03/98.79|28.66/95.19|16.82/97.46|12.76/97.81
> |Ours (P-T-I-S)|0.55/99.77|5.89/98.83|29.61/94.64|17.93/96.76|13.50/97.50
> |Ours (T-I-S-P)|0.51/99.83|4.94/98.95|27.53/95.51|17.40/97.13|12.60/97.86

---

> > ### Author Rebuttal · Reviewer_ABCp · 2026-04-01
> >
> > Thanks for the rebuttal. I will keep my positive score.

---

> > > ### Author Response · Authors · 2026-04-02
> > >
> > > Thank you very much for your response and for confirming that all concerns have been addressed. Thank you again for your time and consideration.

---

### Official Review · Reviewer_wyoZ · 2026-03-08

**Soundness:** 3
**Presentation:** 3
**Significance:** 3
**Originality:** 3
**Overall Recommendation:** 4
**Confidence:** 4

**Summary:**

This paper challenges the "text-as-prototype" paradigm in zero-shot OOD detection with CLIP. The authors theoretically demonstrate an intrinsic "modality gap" where text embeddings are misaligned with optimal visual prototypes. To solve this problem, the author proposes an online pseudo-supervised framework that learns class prototypes directly in the visual feature space using unlabeled test-time data streams and soft predictions from the VLM. Extensive experiments across multiple OOD detection benchmarks demonstrate that consistent improvements over prior zero-shot OOD detection methods

**Compliance With Llm Reviewing Policy:**

Affirmed.

**Final Justification:**

I have checked the rebuttal and I will maintain the positive score.

**Key Questions For Authors:**

1.	In some experiments, training-free approaches appear to perform competitively with or even outperform training-based methods. Could the authors clarify why this might occur?
2.	The prototype updates rely on soft predictions from the pre-trained CLIP. How does the method behave when CLIP performs poorly, for example, on domain-specific datasets such as medical images or other specialized domains?
3.	Since the algorithm updates prototypes in an online manner, the distribution in the input (e.g., covariate shift) may affect the learned prototypes. Did the authors consider scenarios where the test distribution may change over time, and how robust the method is under such conditions?

**Limitations:**

yes

**Strengths And Weaknesses:**

Strengths:

1.	The problem formulation is clear and well-motivated. The paper challenges a widely adopted assumption in CLIP-based OOD detection: the text embeddings of OOD classes can perfectly serve as visual class prototypes. The authors argue that this assumption introduces systematic bias when computing OOD scores, which motivates further analysis and method design.
2.	The theoretical analysis is solid and supports the motivation. The paper provides a formal theoretical analysis to show that the optimal visual prototypes must lie in the subspace spanned by visual features.
3.	The proposed method is intuitive and reasonable. By utilizing online adaptation, the proposed method addresses the modality gap by gradually transforming text-based prototypes into visual prototypes. It leverages pseudo-labels from VLMs to update prototypes incrementally without labeled data.
4.	The empirical validation is extensive. The proposed framework demonstrates superior performance across diverse OOD benchmarks and experimental setups with low standard deviations. In addition, the ablation study in Figure 1 indicates that the method is relatively robust to the choice of hyperparameters

Weaknesses:

1.	The discussion on online prototype adaptation methods is limited. The proposed approach relies on online updates of class prototypes using test-time samples. However, the paper provides limited discussion of related topics like test-time adaptation [1-3], which also updates the class prototype during inference time.
2.	The discussion on inference-time details is insufficient. Since the proposed method performs online updates during inference, the batch size or the order of test samples may influence the stability of prototype updates. It would be better if the paper discussed whether different batch sizes may affect the results.

[1] Iwasawa Y, Matsuo Y. Test-time classifier adjustment module for model-agnostic domain generalization[J]. Advances in Neural Information Processing Systems, 2021, 34: 2427-2440.

[2] Jang M, Chung S Y, Chung H W. Test-time adaptation via self-training with nearest neighbor information[C]. ICLR, 2023.

[3] Wang S, Wang J, Xi H, et al. Optimization-free test-time adaptation for cross-person activity recognition[J]. Proceedings of the ACM on Interactive, Mobile, Wearable and Ubiquitous Technologies, 2024, 7(4): 1-27.

---

> ### Author Rebuttal · Authors · 2026-03-26
>
> # 1. Discussion on related works
> Finding that [1,2,3] make a solid contribution to TTA, we are glad to compare with it in the revised version.
>
> While [1,2,3] also update prototypes online, the overlap is mainly at the level of mechanism, not research question. Prior methods use prototype updates as a practical classifier-adjustment tool for test-time adaptation/domain generalization. Our paper instead begins from a theoretical diagnosis of why text-as-prototype is suboptimal in CLIP-based OOD detection, then derives an online visual-prototype learning framework to address that mismatch.
>
> Thus, the novelty of our method is NOT merely “updating prototypes online,” but why the prototypes must be updated, which prototypes are updated, and how the update is formulated and justified.
>
> # 2. Sensitivity to batch size and order
> As described in Sec. 4.2 and Alg. 1, our method is formulated as an **episodic** procedure, where prototypes are updated online whenever each new test sample arrives. Therefore, our default setting is **batch size=1**, which is also theoretically supported by Theorem 4. Although larger batch sizes could be considered as an implementation variant, having access to a batch of test samples during inference is a **strong and impractical** assumption [a].
>
> We investigate sensitivity to ordering in Table 11 (Appendix E), where, in addition to the standard random shuffle (RS), we consider two extreme cases: “ID First” and “OOD First”. For your convenience, the OOD detection results are reported as follows, where the performance drop of ours under the two extreme cases is significantly marginal.
> |FPR95/AUROC|I|S|P|T|Avg
> |-|-|-|-|-|-|
> |AdaNeg+NegLabel|0.59/99.71|9.50/97.44|34.34/94.55|31.27/94.93|18.92/96.66
> |Ours (ID First)|0.52/99.90|5.12/98.91|29.13/95.07|18.35/96.96|13.28/97.71
> |Ours (OOD First)|0.58/99.89|5.39/98.85|30.75/94.86|18.35/96.90|13.75/97.63
> |Ours (RS)|0.50/99.88|5.00/98.94|28.55/95.17|17.12/97.01|12.79/97.75
>
> [a] ReAct: Out-of-distribution Detection With Rectified Activations, NeurIPS 2021
> # 3. Experiments under continuously changing test distribution
> We kindly note that the robustness of our method under temporal shifts is investigated in Table 10 (Appendix E), where the OOD data distribution evolves sequentially over time, e.g., iNaturalist→SUN→Places→Textures(I-S-P-T). In this case, prototypes are initialized only once at the beginning and then updated continuously throughout the stream. For your convenvience, we provide the results in the below.
> |FPR95/AUROC|I|S|P|T|Avg|
> |-|-|-|-|-|-|
> |AdaNeg+NegLabel|0.59/99.71|9.50/97.44|34.34/94.55|31.27/94.93|18.92/96.66
> |I-S-P-T|0.72/99.69|5.76/98.35|29.01/94.69|17.99/97.00|13.37/97.43
> |S-P-T-I|0.52/99.87|5.03/98.79|28.66/95.19|16.82/97.46|12.76/97.81
> |P-T-I-S|0.55/99.77|5.89/98.83|29.61/94.64|17.93/96.76|13.50/97.50
> |T-I-S-P|0.51/99.83|4.94/98.95|27.53/95.51|17.40/97.13|12.60/97.86
> # 4. Why some training-free methods perform better than training-based ones
> The competitive performance of some training-free approaches (e.g., NegLabel) in our experiments mainly stems from their ability to leverage multimodal representations while expanding the label space with negative labels, thereby introducing richer semantic constraints for separating ID and OOD samples. In contrast, many training-based methods (e.g., ViM, VOS, NPOS, LoCoOp) are restricted to a closed ID class space and/or rely primarily on vision-only features, which can limit generalization in open-world settings; this aligns with prior observations on the importance of cross-modal semantics in CLIP-based OOD detection
>
> That said, we do not claim that training-free methods are universally superior. As evidenced by approaches such as LAPT and NegPrompt, incorporating lightweight training (e.g., few-shot prompt tuning) on top of multimodal representations and negative labels can further improve performance. This suggests that training-based and training-free paradigms are better understood as complementary rather than mutually exclusive.
>
> We will clarify this point in the revision to avoid potential misunderstanding.
>
> # 5. Experiments on medical images
> As per your advice, we additionally conduct experiments on the medical domain, following the same setup as [b]. Specifically, we consider ISIC-18[c] as the ID dataset of interest, and select the PathVQA[d] and PatchCamelyon[e] as OOD dataset. We report OOD detection results as follows:
> |FPR95/AUROC|PathVQA|PatchCamelyon|
> |-|-|-|
> |NegLabel|37.44/94.11|48.07/94.76
> |AdaNeg+NegLabel|26.32/95.25|35.98/96.19
> |Ours|20.04/96.66|29.73/97.43
>
> [b] Textual Training for the Hassle-Free Removal of Unwanted Visual Data : Case Studies on OOD and Hateful Image Detection, NeurIPS 2024
>
> [c] Skin lesion analysis toward melanoma detection 2018: A challenge hosted by the international skin imaging collaboration
>
> [d] PathVQA: 30000+ questions for medical visual question answering.
>
> [e] Rotation equivariant CNNs for digital pathology.

---

> > ### Author Rebuttal · Reviewer_wyoZ · 2026-04-01
> >
> > Thanks for the detailed rebuttal. I will maintain the positive score.

---

> > > ### Author Response · Authors · 2026-04-01
> > >
> > > Thank you very much for your response and for confirming that all concerns have been addressed. Thank you again for your time and consideration.

---

### Official Review · Reviewer_RqnG · 2026-03-12

**Soundness:** 3
**Presentation:** 2
**Significance:** 3
**Originality:** 3
**Overall Recommendation:** 4
**Confidence:** 4

**Summary:**

This paper studies zero-shot, post-hoc OOD detection with pre-trained vision–language models and argues that the common practice of using text embeddings as class prototypes is fundamentally flawed due to a modality gap. To address this, the authors propose an online pseudo-supervised procedure that updates prototypes directly in the visual feature space using unlabeled test streams and soft CLIP predictions, with a convergence result for the optimization. The proposed perspective is interesting.

**Compliance With Llm Reviewing Policy:**

Affirmed.

**Final Justification:**

After the rebuttal, I lean toward acceptance and will maintain my positive score 4.

**Key Questions For Authors:**

- How sensitive is the online prototype update to test-stream composition and ordering? This seems important in a streaming setting

- Can the authors discuss more directly the computational trade-off relative to strong post-hoc baselines, especially on ImageNet-scale evaluation?

- Why fpr95 is not shown for some methods on OpenOOD benchmark in Table 2 and Table 3? You can find on https://docs.google.com/spreadsheets/d/1mTFrO-_STYBRcNMMEmHQrFPQzeg6S8Z2vRA8jawTwBw/edit?gid=1185448723#gid=1185448723 What backbone is used?

**Limitations:**

yes

**Strengths And Weaknesses:**

$\textbf{Strengths:}$

- The modality-gap argument is well motivated, and the move from text prototypes to visual-space prototype calibration is sensible. The method is also practically attractive in that it respects the post-hoc setting and does not require retraining on ID data.

- Empirically, the results are strong on OpenOOD benchmarks, the method consistently improves over prior zero-shot training-free baselines,

- The domain-generalization extension in Table 7 also strengthens the practical relevance.

$\textbf{Weaknesses :}$

- The presentation can be improved for clarity.

- The proposed method performs online prototype adaptation using unlabeled test streams, and is therefore not purely “frozen inference” in the same sense as static scoring rules such as MCM or NegLabel. That said, this point should be framed carefully, since the paper already compares against adaptive baselines such as AdaNeg+NegLabel and explicitly discusses this connection.

- The paper should position itself more carefully relative to prior work on test-time prototype adaptation for VLMs [1,2]


[1] Vision-Language Dual-Pattern Matching for Out-of-Distribution Detection, ECCV 2024

[2] Mitigating the Modality Gap: Few-Shot Out-of-Distribution Detection with Multi-modal Prototypes and Image Bias, WACV 2026

---

> ### Author Rebuttal · Authors · 2026-03-26
>
> # 1. The presentation can be improved for clarity.
> Thanks for your helpful suggestion. We are happy to carefully revise it further to improve readability by adding more intuition around theorems and transitions between theoretical and empirical sections.
> # 2. The method is not purely frozen inference
> Thanks for pointing this out. We agree that our method is not a static frozen-inference method in the strict sense of MCM and NegLabel. However, it remains within the post-hoc/training-free zero-shot regime of prior CLIP-based OOD detection work: it uses **NO** ID training data and does **NOT** fine-tune the backbone. We will revise the paper accordingly and describe both our method and AdaNeg as post-hoc zero-shot test-time adaptation methods to avoid confusion.
> # 3. Comparison with related work
> Finding that [1,2] make a solid contribution to OOD detection, we are glad to compare with it in the revised version.
>
> [1] improves over text-only scoring by adding a visual-pattern matching branch, and [2] further incorporates multi-modal prototypes with few-shot prompt tuning to mitigate image–text misalignment. However, both still treat the problem primarily as one of augmenting or refining text prototypes.
>
> In contrast, we show that the limitation is structural: text-derived prototypes are intrinsically separated from the optimal class representatives in the visual feature space by a non-vanishing modality gap.
> Accordingly, we abandon the text-as-prototype assumption itself and directly learn calibrated visual prototypes online from unlabeled test-time streams.
>
> Compared with [1] which is training-free but **requires ID training data**, ours is training-free and zero-shot. Even with such simplicity, the below shows that ours outperforms [1]. (ours is reimplemented on the same pre-trained CLIP model (OpenCLIP B/32) as [1])
> |FPR95/AUROC|iNaturalist (I)|Sun (S)|Places (P)|Textures (T)|Avg
> |-|-|-|-|-|-
> |[1]|12.89/96.94|31.63/92.62|41.15/89.97|32.71/91.60|29.59/92.78
> |Ours |0.46/99.88|4.46/99.10|26.47/86.46|19.01/96.87|12.60/98.08
>
> Compared with [2] which **is  training-required and requires ID training data**, ours is training-free and zero-shot. Even with such simplicity, the below shows that ours outperforms [2].
> |FPR95/AUROC|I|S|P|T|Avg
> |-|-|-|-|-|-
> |[2]|5.91/98.73|15.85/96.08|26.18/93.19|34.88/91.93|20.70/95.38
> |Ours |0.50/99.88|5.00/98.94|28.55/95.17|17.12/97.01|12.79/97.75
> # 4. Sensitivity to composition and ordering
> We address sensitivity to composition in a stronger temporal form in Table 10 (Appendix E). We consider non-stationary test streams where OOD environment changes sequentially over time, e.g., iNaturalist→SUN→Places→Textures(I-S-P-T). In this case, prototypes are initialized only once at the beginning and then updated continuously throughout the stream. The OOD detection results in the below implies the robustness of ours to composition.
> |FPR95/AUROC|I|S|P|T|Avg
> |-|-|-|-|-|-
> |AdaNeg+NegLabel|0.59/99.71|9.50/97.44|34.34/94.55|31.27/94.93|18.92/96.66
> |I-S-P-T|0.72/99.69|5.76/98.35|29.01/94.69|17.99/97.00|13.37/97.43
> |S-P-T-I|0.52/99.87|5.03/98.79|28.66/95.19|16.82/97.46|12.76/97.81
> |P-T-I-S|0.55/99.77|5.89/98.83|29.61/94.64|17.93/96.76|13.50/97.50
> |T-I-S-P|0.51/99.83|4.94/98.95|27.53/95.51|17.40/97.13|12.60/97.86
>
> We investigate sensitivity to ordering in Table 11 (Appendix E), where, in addition to the standard random shuffle (RS), we consider two extreme cases: “ID First” and “OOD First”. For your convenience, the OOD detection results are reported as follows, where the performance drop of ours under the two extreme cases is significantly marginal.
> |FPR95/AUROC|I|S|P|T|Avg
> |-|-|-|-|-|-|
> |AdaNeg+NegLabel|0.59/99.71|9.50/97.44|34.34/94.55|31.27/94.93|18.92/96.66
> |Ours (ID First)|0.52/99.90|5.12/98.91|29.13/95.07|18.35/96.96|13.28/97.71
> |Ours (OOD First)|0.58/99.89|5.39/98.85|30.75/94.86|18.35/96.90|13.75/97.63
> |Ours (RS)|0.50/99.88|5.00/98.94|28.55/95.17|17.12/97.01|12.79/97.75
> # 5. Computational trade-off
> Compared with NegLabel, our method requires online prototype updating during test time, which incurs additional computational overhead. However, the significant performance improvements over NegLabel proves the value of the proposed updating strategy. Besides, our method achieves the best FPR95 and AUROC while requiring substantially less computation than NegLabel+AdaNeg, implies a better trade-off between effectiveness and efficiency.
> ||Computation (by second)|FPR95|AUROC
> |-|-|-|-
> |NegLabel|32.18|25.40|94.21
> |NegLabel+AdaNeg|193.02|18.92|96.66
> |Ours|107.06|12.79|97.75
> # 6. Clarity on Tables 2 and 3
> The baseline results in Tables 2 and 3 were directly copied from AdaNeg, where, same as AdaNeg, the backbone is CLIP B/16. Therefore, the missing FPR95 entries were inherited from AdaNeg’s original presentation **NOT** intentionally omitted by us. Based on the provided link, we will supplement the missing results in the revised paper.

---

> > ### Author Rebuttal · Reviewer_RqnG · 2026-04-01
> >
> > Thanks for the detailed rebuttal. After carefully considering the response and the reviews from other reviewers, I lean toward acceptance and will maintain my positive score.

---

> > > ### Author Response · Authors · 2026-04-02
> > >
> > > Thank you very much for your response and for confirming that all concerns have been addressed. Thank you again for your time and consideration.

---

### Decision · Program_Chairs · 2026-04-30

**Decision:**

Accept (regular)

**Comment:**

This paper investigates the non-monotonic relationship between feature-space geometry and OOD detection using a layer-wise geometric regularizer called CIALLO as a probing tool, showing that geometric improvements benefit OOD performance only in moderate tasks and become ineffective in hard, rigid representation regimes. The authors are recommended to add additional results into the paper to further improve its quality.